

# Modelling the radiative effects of smoke aerosols on carbon fluxes in Amazon

Demerval S. Moreira[1,2], Karla M. Longo[3,a], Saulo R. Freitas[3,a], Marcia A. Yamasoe[4], Lina M. Mercado[5,6], Nilton E. Rosário[7], Emauel Gloor[8], Rosane S. M. Viana[9], John B. Miller[10], Luciana V. Gatti[11,12], Kenia T. Wiedemann[13], Lucas K. G. Domingues[11,12], and Caio C. S. Correia[11,12]

[1]Universidade Estadual Paulista (Unesp), Faculdade de Ciências, Bauru, SP, Brazil.
[2]Centro de Meteorologia de Bauru (IPMet), Bauru, SP, Brazil.
[3]Centro de Previsão de Tempo e Estudos Climáticos, Instituto Nacional de Pesquisas Espaciais (INPE), Cachoeira Paulista, SP, Brazil.
[4]Departamento de Ciências Atmosféricas do Instituto de Astronomia, Geofísica e Ciências Atmosféricas, Universidade de *São Paulo* (USP), *São Paulo*, SP, Brazil.
[5]Geography, College of Life and Environmental Sciences, University of Exeter, Exeter, UK.
[6]Centre for Ecology and Hydrology (CEH), Wallingford, UK.
[7]Universidade Federal de *São Paulo* (UNIFESP), Campus Diadema, Diadema, SP, Brasil.
[8]School of Geography, University of Leeds, Woodhouse Lane, Leeds, UK.
[9]Departamento de Estatística, Universidade Federal de Viçosa (UFV), Viçosa, MG, Brazil.
[10]Global Monitoring Division, Earth System Research Laboratory, National Oceanic and Atmospheric Administration (NOAA), Boulder, Colorado 80305, USA.
[11]Centro de Ciências do Sistema Terrestre, Instituto Nacional de Pesquisas Espaciais (INPE), São José dos Campos, SP, Brazil.
[12]Instituto de Pesquisas Energéticas e Nucleares (IPEN) – Comissão Nacional de Energia Nuclear (CNEN), São Paulo, Brazil.
[13]Department of Ecology and Evolutionary Biology, University of Arizona, Tucson, AZ, USA.
[a]Now at Universities Space Research Association/Goddard Earth Sciences Technology and Research (USRA/GESTAR) at Global Modeling and Assimilation Office, NASA Goddard Space Flight Center, Greenbelt, MD, USA.

*Correspondence to:* Demerval S. Moreira (demerval@fc.unesp.br)

**Abstract.** Every year, a dense smoke haze of regional dimensions covers a large portion of South America originated from fire activities in the Amazon Basin and Central parts of Brazil during the dry/biomass-burning season between August and October. Over a large portion of South America, the average aerosol optical depth at 550 nm exceeds 1.0 during the fire season while the background value during the rainy season is below 0.2. Smoke aerosol particles increase scattering and absorption of the incident solar radiation. The regional-scale aerosol layer reduces the amount of solar energy reaching the surface, cools the near surface air, and increases the diffuse radiation fraction over a large disturbed area of the Amazon rainforest. These factors affect the energy and $CO_2$ fluxes at the surface. In this work, we applied a fully integrated atmospheric model to assess the impact of smoke aerosols in $CO_2$ fluxes in the Amazon region during 2010. We address the effects of the attenuation of the solar global radiation and the enhancement of the diffuse solar radiation flux inside the canopy. Our results indicated that the smoke aerosols led to an increase of about 22% of the gross primary productivity of Amazonia, 9% of plant respiration and a decline in soil respiration from of 3%. Consequently, Amazonia net ecosystem exchange during September 2010 dropped from +101 to -104 TgC when the aerosol effects were considered, mainly due to the aerosol diffuse radiation effect. For the





forest biome, our results pointed to a dominance of the diffuse radiation effect on $CO_2$ fluxes, reaching a balance of 50% – 50% between the diffuse and direct aerosol effects for high aerosol loads. For C3 grass type and *cerrado*, as expected, the contribution of the diffuse radiation effect is much lower, tending to zero with the increase of aerosol load. That is, the Amazon during the dry season, in the presence of high smoke aerosol loads, change from being a source to be a sink of $CO_2$ to the atmosphere.

# 1 Introduction

The austral winter in most of South America is typically dry with extensive vegetation fires, mostly human induced, in areas of deforestation and agricultural/pasture land management. The fire activity, especially in Amazonia and *cerrado* (a savanna-like biome), usually last for at least 3 months, from August to October every year, and has been typically called "biomass burning season". The fire emissions are an additional source of carbon dioxide ($CO_2$), and a significant source of several other trace gases to the atmosphere (Andreae et al., 2002). In addition to trace gases, vegetation fires also produce a significant amount of aerosol particles, in particular from the fine mode, which, on average, contribute to at least 90% of the total AOD in the visible spectrum in the case of the South America regional smoke (Reid et al., 2005; Rosário, 2011). The resulting smoke haze has regional dimensions over South America covering areas of about several millions of $km^2$ (Prins et al., 1998). Out of the biomass burning season, mean AOD in the visible spectrum varies from 0.10 to 0.15 across the Amazon rainforest, and from 0.12 to 0.20 in the *cerrado* areas (Schafer et al., 2008; Rosario, 2011). At the peak of the burning season, during September, monthly mean AOD can reach values up to 10 times in the southern portion and up to 5 times, in the northern areas, compared to those observed during clean conditions (Schafer et al., 2008; Rosário, 2011). Ångström exponent (AE), an optical property applied to characterize aerosol particle size based on the spectral dependence of AOD, increases from 0.60, during the clean period, to 2.0 during the peak of the burning season (Schafer et al., 2008). Such high value of AE is representative of air masses dominated by fine mode particles, a major feature of the south America regional plume. Low values of AE indicate a dominance of coarse mode particles, a characteristic of the pristine region of the Amazon dominated by biogenic particles. The particles absorption characteristics, expressed in optical property Single Scattering Albedo (SSA), is critical to the smoke plume interaction with radiation. SSA range in the southern portion of Amazon forest during burning season, 0.92-0.93 at 550 nm, reveals a dominance of moderate absorbing particles (Schafer et al., 2008), unlike those from *cerrado*, which are highly absorbing and present low value of SSA ( 0.89±0.04). The *cerrado* substantial lower SSA than the mean values in the southern Amazon forest is due to the higher fraction of flaming phase combustion, typical of savanna-like vegetation (Schafer et al., 2008).

In Amazonia under heavy smoke conditions, the surface cooling can reach 3° C, restraining the turbulent flows and consequently the evapotranspiration of water and sensible heat flux. Thus, the result is a drier/shallower boundary layer, inhibiting convective clouds formation/development and hence decreasing precipitation (Yu et al., 2002). In particular, smoke reduces the net direct solar radiation reaching the surface while it increases the diffuse fraction of solar radiation. The diffuse fraction of the photosynthetically active radiation (PAR) can increase from about 19%, which is the typical value for a clean atmosphere





scenario, up to 80% under smoky conditions (Yamasoe, et al., 2006). Smoke aerosols also act as cloud condensation nuclei affecting cloud microphysics properties and therefore, changing the radiation budget and hydrological cycle over disturbed areas (Kaufman, 1995; Rosenfeld, 1999; Andreae, et al., 2004; Koren et al., 2004).

Changes in the total downward solar irradiance at the surface usually does not impact the photosynthetic activity of the leaves

on the top of the forest canopy because those are usually light saturated around midday, closing the plants' stomata. On the other hand, sub-canopy leaves remain typically under light-deficit conditions and do not fully deplete their photosynthetic potential. Then, increasing the diffuse light, which penetrates deeper into the canopy, increases PAR availability to the sub-canopy leaves and the rate of photosynthesis, and, consequently, the atmospheric carbon assimilation (Baldocchi, 1997; Misson et al., 2005; Oliveira et al., 2007; Knohl and Baldocchi, 2008; Mercado et al., 2009; Doughty et al., 2010; Kanniah et al., 2012; Rap et al.,

2015). However, the increase of the diffuse radiation is also accompanied by a decrease in the net radiation, therefore defining an optimal diffuse to total radiation fraction that allows a maximum of carbon assimilation. Under heavy pollution or cloudy sky, plant productivity enhances with the increase of the diffuse radiation, though it is still insufficient to compensate the reduction of the total irradiance reaching the surface. Additionally, other authors suggest that the contribution of smoke aerosols on $CO_2$ assimilation can also be due to the cooling of the air (Min, 2005; Doughty et al., 2010; Steiner and Chameides, 2011), causing

an increase of relative humidity (Collatz et al., 1991) near the earth's surface, which reduces respiration and thermal stress of the leaves. Indeed, field observations indicate that tropical forest productivity is highly sensitive to temperature variations (Feeley et al., 2007), with $CO_2$ assimilation decreasing sharply during warmer periods. Doughty et al. (2010) estimated that, under dense smoky conditions in Amazonia, 80% of the increase of $CO_2$ absorption is due to the increase of sub-canopy light (diffuse radiation), and only 20% is due to the reduction of the canopy temperature. A modeling study at the global scale

exploring the aerosol impact on gross primary productivity ($GPP$) concluded that the positive effect of the diffuse radiation increase was indeed larger than the negative effect of the irradiance reduction (Mercado et al., 2009). JULES simulations forced with aerosol fields from the Hadley Centre Global Environment Model version 2 (HadGEM2) made by these authors pointed to an increase in the diffuse fraction of irradiance, and a consequent increase in the global carbon uptake of 23.7% from 1960 to 1999. Rap et al. (2015) also using JULES, but with a different off-line aerosol model – 3-D GLObal Model of Aerosol

Processes (GLOMAP, Mann et al., 2010), estimated that the smoke aerosols affected the diffuse radiation by 3.4 to 6.8% and increased the net primary production ($NPP$) of 1.4 to 2.8% in Amazonia during the period between 1998 and 2007. Besides, the smoke aerosol indirect effect will also impact the $CO_2$ fluxes by changing the amount of rain (and soil moisture), and solar radiation availability and diffuse fraction. Lastly, vegetation fires also emit ozone precursors and promote tropospheric ozone production, and surface deposition. Ozone is highly phytotoxic, damaging plants stomata and reducing $CO_2$ uptake (Sitch

et al., 2007).

Motivated by all these previously cited recent observational and theoretical studies that have demonstrated the impacts of aerosols on the $CO_2$ fluxes, we applied an integrated in-line numerical atmospheric modeling system to explore the following scientific questions: what is the relative role of the main processes between soil/vegetation and the atmosphere controlling the carbon cycle in Amazonia? What are the effects of smoke aerosols for each of these processes? What is the net aerosol effect

on the $CO_2$ fluxes? What is the relative effect of the direct interaction of aerosol radiation against the aerosol impact on the





diffuse fraction? What is the regional dimension of the aerosol impact on $CO_2$ fluxes? Finally, how well can a state-of-the-art Chemical Transport Model (CTM) reproduce $CO_2$ fluxes and mixing ratio observations over the Amazon Basin?

The structure of this paper is as follows. In Sect. 2.1, we present a description of the most relevant aspects of the integrated atmospheric modeling system for this application. The adopted model configuration and the input data sets, including emis-
sions and the ones used as boundary conditions, are described in Sect. 2.2. The observational datasets, both from direct and remote sensing observation, used for model evaluation and analyses are described in Sect. 2.3. Model results are presented and discussed in Sect. 3. We start in Sect 3.1, providing an overview of the meteorological scenario and fire activity in Amazonia during the study period, including both model results and observation. We then follow up in Sect. 3.2 with the model results for the smoke regional plume. Finally, in Sect. 3.3 we examine the model results for energy and $CO_2$ fluxes, related to several
surface/atmosphere interaction processes, as well as the smoke aerosol impacts on them. The main results are summarized in Sect. 4.

## 2    Methods and data sets

### 2.1    Description of the relevant parts of the modeling system

In this work, we employed the integrated atmospheric-chemistry model BRAMS version 5.0 (Brazilian developments on the
Regional Atmospheric Modeling System, Freitas et al., 2005, 2009, and 2016), which has been coupled in a two-way mode with the Joint UK Land Environment Simulator v3.0 (JULES), the land surface scheme of the UK Hadley Centre Earth System model (Moreira et al., 2013). JULES simulates the exchange of carbon, momentum, and energy between the land surface and the atmosphere. Additionally, it represents sub-surface hydrological processes, plants photosynthesis and respiration, and vegetation and soil dynamics (Best et al., 2011; Clark et al., 2011). The photosynthesis-radiation scheme, accounts for the
effects of diffuse radiation on canopy photosynthesis, scaling photosynthesis from the leaf to the canopy, using a multilayer approach to simulate radiation interception and photosynthesis (Mercado et al., 2009; Dai et al., 2004). JULES separates direct and diffuse radiation and sunlit and shaded leaf photosynthesis on each layer. Sunlit leaves receive both direct and diffuse light while shaded leaves only receive diffuse radiation (Clark et al 2011). Evaluation of the skill of JULES in simulating GPP under high direct and high diffuse radiation conditions has been tested against flux sites in the Amazon and in temperate forest
sites where direct and diffuse radiation measurements are available. This is shown in Figure 2 of Rap et al. (2015) at Tapajos and French Guyana in the Amazon and in Mercado et al. (2009) (Figure 1) for two temperate forest sites. BRAMS is also in-line coupled with a Eulerian transport model (CCATT) suitable to simulate transport, dispersion, chemical transformation and removal processes associated with trace gases and aerosols (Longo et al., 2013). In CCATT, aerosol and trace gas transport runs consistently in-line with the atmospheric state evolution using the BRAMS dynamic and physical parameterizations. The tracer
mass mixing ratio, which is a prognostic variable, includes the effects of sub-grid scale turbulence in the planetary boundary layer, convective transport by shallow and deep moist convection in addition to grid scale advection transport. For the gaseous species, CCATT-BRAMS in principle can employ several gaseous chemical mechanisms. However, for this study, only carbon monoxide (CO), $CO_2$ and aerosol particles (smoke type) were emitted and transported. The physical removal processes (dry




and wet deposition) were applied to all the three tracers, and effective lifetimes were applied to CO and aerosol particles. The modeling of smoke aerosol particles is the focus of the present study, therefore only biomass burning emission sources were considered.

The BRAMS model parameterizations chosen for the simulations performed in this work are described as follows. The
parameterization for the unresolved turbulence in the planetary boundary layer (PBL) was based on the Mellor and Yamada (1982) formulation, which predicts turbulent kinetic energy (TKE). For the microphysics, we used the single-moment bulk microphysics parameterization, which includes cloud water, rain, pristine ice, snow, aggregates, graupel and hail (Walko et al., 1995). It includes prognostic equations for the mixing ratios of rain and each ice category of total water and the concentration of pristine ice. Water vapor and cloud liquid mixing ratios are diagnosed from the prognostic variables using the saturation mixing
ratio with respect to liquid water. The deep and shallow cumulus convection schemes are based on the mass-flux approach and described in Grell and Freitas (2014).

The radiation scheme is a modified version of the Community Aerosol and Radiation Model for Atmosphere (CARMA, Toon et al., 1988), which includes the aerosol-radiation interaction with feedback to the model heating rates (Longo et al., 2013, Rosário et al., 2013). In addition, we included in CARMA a parameterization to calculate the diffuse fraction of solar irradiance
specific for smoke aerosols in Amazonia. This parameterization was based on measurements of broad and narrowband solar global and diffuse irradiance components performed with a Multi-filter Rotating Shadow-band Radiometer (MFRSR – Harrison et al, 1994). With the narrowband measurements, centered at 415, 670, 870 and 1036 nm, AOD was estimated following the methodology of Harrison and Michalsky (1994) and Rosário et al. (2008). The measurements were performed at Reserva Biológica do Jaru, RO (-10.145°, -61.908°) during the dry/smoke season of 2007. The diffuse fraction of broadband irradiance
reaching the surface as a function of AOD at 670 nm wavelength for distinct optical air mass intervals (m), expressed as a ratio of the optical path length relative to the path length at zenith, (Figure 1) was analyzed and a three-degree polynomial fit was proposed as follows:

$$D = aAOD_{670}^3 + bAOD_{670}^2 + cAOD_{670} + d \qquad (1)$$

The values of the fitting parameters $a$, $b$, $c$, and $d$, of the function in Equation 1 are described in Table 1. These fittings were
achieved after filtering the data for clouds, which can be present even during the dry season, especially during days with low AOD values. When clouds are present, the diffuse fraction of radiation increases significantly with values as high as a very polluted atmosphere. However, as discussed below, the analysis presented here focuses only in areas/hours without cloud cover, i.e. the results were obtained by filtering out the points with cloudiness.

## 2.2   Model configuration and input data sets

BRAMS model simulations were conducted for a domain covering the northern part of South America (southwest corner: 18° S, 90° W, northeast corner: 15° N, 30° W) using a regular grid with 20 km resolution, as illustrated in Figure 2. JULES was configured with 10 canopy layers. The chosen model domain encompasses the Legal Brazilian Amazon region (LBAR, depicted by the red line in Figure 2), which is a region of 5,016,136.3 km² (59% of the Brazilian territory). It hosts approximately



24 million inhabitants, which is only around 12% of the Brazilian population. The Brazilian Federal Law Nº 5.173 (Art. 2) established LBAR in 1966 as an administrative unit to promote sustainable development in one of the most, if not the most, rich region in the Brazilian territory in terms of natural resources. The main tropical biomes in South America, the Amazon rainforest, *Cerrado* and Pantanal (wetlands), are all found in LBAR. Despite government protection, deforestation

activities, followed by vegetation fires, have led to extensive land use change to pasture and agricultural fields, and changes in atmospheric aerosol load and characteristics. This study aims to analyze the effect of these changes on the atmospheric environment, radiation budget, and forest productivity in this important region.

The NCEP Global Forecast System analysis (http://rda.ucar.edu/datasets/ds083.2/), with 6-hourly time resolution and 1° × 1° spatial resolution, provided initial and boundary conditions for the time integration of the meteorological fields. Sea

surface temperature (SST) was taken from NOAA Optimum Interpolation (OI) SST product, version 2, with 1° × 1° spatial resolution (available at http://www.esrl.noaa.gov/psd/data/gridded/data.noaa.oisst.v2.html, Reynolds et al., 2002). Soil moisture was initialized with the soil moisture estimation operational product developed by Gevaerd and Freitas (2006) and available at CPTEC/INPE, and the soil temperature was initialized assuming a vertically homogeneous field defined by the air temperature closest to the surface from the initial atmospheric data. The carbon data assimilation system, Car-

bon Tracker 2015, (Krol et al., 2005; Peters et al., 2007), with 3° × 2° spatial resolution and 34 vertical levels, (available at http://www.esrl.noaa.gov/gmd/ccgg/carbontracker/), provided $CO_2$ initial and boundary conditions. Initial and boundary conditions for carbon monoxide (CO) were based on optimized fluxes, with 1° × 1° spatial resolution, as calculated by the 4D-var system using the Infrared Atmospheric Sounding Interferometer (IASI) data taken onboard the Eumetsat Polar System (EPS) Metop-A Satellite (Krol et al., 2013).

Biomass burning emissions of trace gases and aerosols were provided by the Brazilian Biomass Burning Emission Model (3BEM, Longo et al., 2010). The 3BEM emissions are based on a database of fire pixel counts and burned area derived from the combination of remote-sensing fire products from Geostationary Operational Environmental Satellite-Wildfire Automated Biomass Burning Algorithm (GOES WF-ABBA product; Prins et al., 1998), the Brazilian National Institute for Space Research (INPE) fire product, which is based on the Advanced Very High Resolution Radiometer (AVHRR) aboard the NOAA polar

orbiting satellites series (Setzer and Pereira, 1991), and the Moderate Resolution Imaging Spectroradiometer (MODIS) fire product (Giglio et al., 2003). Fire emissions were split into smoldering and flaming emission contributions, releasing trace gases and aerosol particles in the lowest model layer and in the injection layer, respectively, as determined by the in-line plume rise model (Freitas et al., 2007, 2010).

The land use dataset from the United States Geological Survey (USGS) at 1-km resolution was merged with a land cover map

for the Brazilian legal Amazon region (PROVEG) (Sestini et al., 2003). PROVEG is based on the Landsat Thematic Mapper (TM) images with spatial resolution of 90 m × 90 m from the year 2000 and deforestation data from the Amazon Deforestation Monitoring Program (PRODES) for the year 1997. At the 20-km resolution, each grid-box has 400 specifications of vegetation, which was reduced to 9 patches of different land cover categories (broadleaf trees, needle leaf trees, C3 and C4 grasses, shrubs, urban, inland water, soil and ice), each with its respective occupation fraction. JULES treats each category separately and

returns to BRAMS average fluxes weighted by the occupation fraction. The model results are then discussed for the land cover





categories present in the considered model domain: broad leaf trees (tropical forest), shrubs (*cerrado*), and C3 e C4 grasses types (pasture). The land use map in the model domain is illustrated in Figure 2.

The model simulations were initialized on 15 August 2010 00:00 UTC and conducted for 45 days, with the analysis of model results restricted to the month of September only to avoid model spin-up artifacts.

5    A set of three experiments was performed. In the first one (hereafter named NO-AER), the aerosol-radiation interaction was neglected. The direct aerosol effect was taken into account in both the second (hereafter named DIR-AER) and third (hereafter named DIR+DIF) experiments, but only in the latter one the diffuse fraction of radiation was passed to JULES, otherwise it was set zero. Additionally, a long-term model run (2 years, from January 2010 to December 2011) was carried out only for the DIR+DIF model configuration.

## 2.3    Method of analysis of the model results

The carbon fluxes from the three model simulations should allow assessing the aerosol effect on $CO_2$ uptake in Amazonia and the relative role of surface temperature due to aerosol direct effect and the increase in the diffuse fraction of PAR due to aerosol scattering. The $CO_2$ fluxes from the model was analyzed as the variation related to the total aerosol effect (both on diffuse radiation and direct radiation):

$$\Delta Flux_{tot} = Flux_{DIR+DIF} - Flux_{NO-AER} \tag{2}$$

Only with the direct radiation aerosol effect:

$$\Delta Flux_{dir} = Flux_{DIR-AER} - Flux_{NO-AER} \tag{3}$$

And, only with the diffuse radiation aerosol effect:

$$\Delta Flux_{diff} = Flux_{DIR+DIF} - Flux_{DIR-AER} \tag{4}$$

20    We examined the spatial distribution and diurnal cycles of $CO_2$ fluxes, related to several surface/atmosphere interaction processes: The Gross Primary Production ($GPP$) is the total carbon uptake in the process of photosynthesis by plants, especially leaves, in an ecosystem over a land area. As so, $GPP$ is essentially a response to the amount of photosynthetically active radiation portion of solar energy reaching the plants. In addition, we also looked at the $CO_2$ fluxes associated with plant respiration ($R_P$), soil heterotrophic respiration ( $R_P$,), and the net ecosystem exchange ($NEE = R_P + R_H - GPP$), which is a measurement of the quantity of carbon entering and leaving the ecosystem (positive when the ecosystem is a $CO_2$ sink, and negative when is a $CO_2$ source). The spatial distributions of $CO_2$ fluxes are presented as monthly mean in $\mu molC\,m^{-2}\,s^{-1}$ for September 2010 at 1600 UTC, which is around noon local time. The diurnal cycles of $CO_2$ fluxes are presented hourly as monthly mean in $\mu molC\,m^{-2}\,s^{-1}$ for September 2010, considering the model grid-cells circumscribed in the LBAR region delimited in Figure 2. The monthly net values of $CO_2$ fluxes in the LBAR are calculated as the integral of the mean diurnal cycles, and presented in $TgCmonth^{-1}$, considering the LBAR area $\sim 5.2\ 10^{12}\ m^2$, which is the total area of the model cells circumscribed in the LBAR region delimited in the Figure 2, multiplied by 30 to get the monthly total.



As each biome has its own characteristics and responds differently to changes occurring on the surface and in the atmosphere, we first examined how each biome responds to the presence of smoke aerosol, both in terms of total irradiance attenuation near surface and increase of the diffuse fraction of PAR. We also evaluated the relative contribution of the diffuse to the total (diffuse + direct) aerosol effect on the $CO_2$ fluxes for each the biome type. The contribution of the direct and diffuse radiation effect to $CO_2$ fluxes were, respectively, defined as:

$$Flux_{\%dir} = \frac{\Delta Flux_{dir}}{\Delta Flux_{tot}} \tag{5}$$

$$Flux_{\%diff} = \frac{\Delta Flux_{diff}}{\Delta Flux_{tot}} \tag{6}$$

where $\Delta Flux_{tot}$, $\Delta Flux_{dir}$ and $\Delta Flux_{diff}$, follow the definitions in Eq. 2, 3, and 4, respectively. After assessing the specific behavior of each biome, model results for $CO_2$ and energy fluxes from each biome were then averaged, weighting by the biome fraction of each grid, to address the heterogeneity of the Amazon region in terms of land cover and local climates.

## 2.4 Data sets for model evaluation

The model results for precipitation and near-surface temperature were contrasted with direct observations and products derived from satellite observations. The smoke aerosol spatial distribution from the model was validated with remote sensing products. Additionally, model performance on simulating CO and $CO_2$ mixing ratios was assessed using measurements of carbon monoxide (CO) and $CO_2$ concentration in air samples collected over the Amazon during 2010 and 2011. Biomass burning mainly releases water vapor and $CO_2$ to the atmosphere but is also a major source of other tracers, such as CO, volatile organic compounds, nitrogen oxides, and organic halogen compounds (Andreae and Merlet, 2001). An enhancement of CO has been historically observed in Amazonia during the dry season, which is mostly attributed to fire emissions because volatile organic compounds (VOC) oxidation has very little seasonality over there (Holloway et al., 2000; Duncan and Logan, 2008, Andreae et al., 2012). Therefore, the evaluation of CO mixing ratio from the model against observation provides an assessment of model skills to simulate fire emission, its transport, and removal. We also looked into fire activity and used the soil moisture, and meteorological variables from the model as indicators of spatial scale of locations where fires were more likely to occur. The datasets used for model evaluation are described below.

**Precipitation:** Monthly mean precipitation over the Amazon region was obtained from the algorithm 3B42 of the Tropical Rainfall Measuring Mission (TRMM) merged high quality (HQ)/infrared (IR) precipitation product at a spatial resolution of $0.25° \times 0.25°$ (http://trmm.gsfc.nasa.gov/3b42.html; Kummerow et al., 1998; Kawanishi et al., 2000). TRMM 3B43 is derived from retrievals of 3-hourly precipitation amount from the precipitation radar (PR), TRMM microwave imager (TMI), and visible and infrared scanner (VIRS) aboard the TRMM satellite, merged with rain gauge data from the Climate Anomaly Monitoring System (CAMS) and the Global Precipitation Climatology Project (GPCP). Satellite estimates of precipitation were used for model evaluation due to their more complete spatial and temporal coverage compared to rain gauge data. The





latter were also complementarily used to evaluate modelled precipitation and temperature, though not ignoring the low density and heterogeneous distribution of the observational network in the geographical model domain: 72 PCDs (automatic stations installed and maintained by the Brazilian National Institute of Meteorology - INMET) in 5,016,136.3 $km^2$.

**Temperature:** We evaluated the mean diurnal cycle of 2-meter temperature from the model with data from 72 near-surface measurement ground stations in the LBAR, locations depicted in Figure 3.a.

**Fire activity:** We checked the coherence of soil moisture results from the model with the fire product from Advanced Very High Resolution Radiometer (AVHRR) onboard the NOAA polar orbiting satellites series. The fire detection used is based on

the AVHRR retrieval algorithm from the Brazilian National Institute for Space Research (www.cptec.inpe.br/queimadas).

**Smoke CO and $CO_2$:** Model performance on simulating CO and $CO_2$ mixing ratios was assessed using measurements of carbon monoxide (CO) and $CO_2$ concentration in air samples collected over the Amazon during 2010 and 2011. The air samples were collected with portable sampling systems consisting of separate compressor and flask units (Tans et al., 1996)

onboard a Cessna 206 aircraft in descending spirals from 4,300 to 300 m over four Amazon locations indicated in Figure 2: *Santarém*, PA (2.43° S, 54.72° W), *Rio Branco*, AC (9.97° S, 67.81° W), *Alta Floresta*, MT (12.54° S, 55.71° W), and *Tabatinga*, AM (4.25° S, 69.94° W). The air samples collected in *Santarém*, *Rio Branco* and *Alta Floresta* are characteristic of the moist tropical forest, both primary and secondary, surrounding them. During the dry season, both local (forest), and remote *cerrado* and pasture fire emissions affect these three sites. The samples collected in *Tabatinga*, which is further west, in a more

pristine area of the Amazon forest, respond to the influence of the intact forest landscape upwind. During the dry season, fire emissions influence the atmospheric chemistry in all sites, although the smoke impact in *Tabatinga* is more related to episodic long-range transport events, and so not as persistent as the others. The air sampling was always between 12:00 and 14:00 local time and analyzed at the laboratory of the *Instituto de s Energéticas e Nucleares* (IPEN) in *São Paulo*, Brazil. Measurements precision of $CO_2$ and CO is estimated to be around 0.04 ppmv and 0.5 ppbv, respectively. For further details, regarding air

sampling and analytical methods, see Gatti et al. (2010).

We also used $CO_2$ measurements in the *Tapajós* National Forest near km 67 (02.85°S, 55.04°W) of the *Santarém* – Cuiabá highway, just south of *Santarém*, for model evaluation. The *Tapajós* measurements are based on eddy covariance methods, using the profile mixing ratio data to estimate the change in vertical average mixing ratio between the ground and flux measurement height to calculate the column average storage of $CO_2$ (Saleska et al., 2003). $CO_2$ mixing ratio was measured at

8 levels along the tower (62.2, 50, 39.4, 28.7, 19.6, 10.4, and 0.91 m). Sample air was drawn and analyzed with an infrared gas analyzer (IRGA, LI-6262, Licor, Lincoln, NE). Pressure and temperature of the Licor cells are controlled. Eddy Licors are automatically zeroed every 2 hours and the Licor profile, every 20 minutes. All Licors are automatically calibrated with span gases every 6 hours.

**Smoke aerosol:** The AOD (Aerosol Optical Depth) product derived from MODIS sensors onboard of the Aqua satellite is





used to evaluate the simulated smoke aerosol plume. In this work, we used the MODIS Level 2.0 Collection 5.1 (051) data and Level 3 Atmospheric product denominated MYD08_D3 (Mean Aerosol Optical Thickness at 550 nm).

## 3 Model results and discussion

### 3.1 Meteorology, fire activity and regional smoke plume

**Temperature:** On a monthly mean basis, the simulated 2-meter temperature in the central portion of the Amazon Basin during September 2010 peaked around 32 °C at 1600 UTC, while the northwest region was slightly cooler, typically ranging from 30 to 28 °C (Figure 3.a). Going southeast, towards the *cerrado* region, the mean 2-meter temperature reaches 35 °C at 1600 UTC. The temperature gradient is mostly associated with the gradient of soil moisture and land cover in the region. The evaluation of the mean diurnal cycle of 2-meter temperature from the model against observations using 72 near-surface measurement ground stations in the LBAR during the same period shows that the model results are consistent with observations (Figure 3.b), though the model mean temperature was typically cooler ( 2.5 °C) during the night period but still within the standard deviation of the mean temperature observed. Also, the model temperature has a diurnal cycle early in 1-hour with respect to observation.

**Precipitation:** The monthly mean rainfall data from the ground stations monitoring network, interpolated to the model grid points (Figure 4.a), and the TRMM rainfall product (Figure 4.b), both reveal a well-defined spatial distribution of the precipitation in the northern region of South America during September 2010. First, in the Southeast Amazon region accumulated rainfall was low, with values typically lower than 50 mm for this month; on the other side, the northwestern region was relatively wetter, with accumulated values around 200 mm. In addition, there were few cores of large rainfall in the northern part of South America, mainly in the Guiana Highlands, and associated with the topography forcing. The general spatial distribution of model accumulated precipitation (Figure 4.c) compares well with ground observation (Figure 4.a), though with an indication that the model overestimated the total precipitation. However, one must take into account that the measurement stations are very scarce in this region and several precipitating systems might have been missed. Interpolation in the presence of limited information usually reduces the intensity of precipitation, spreading the value observed around the neighboring grid points without data available. Indeed, the precipitation estimated by TRMM (Figure 4.b) is in a much better agreement with the model results (Figure 4.c).

**Soil moisture and fire activity:** As a result of the precipitation distribution, the soils are predominantly wetter in the northwestern part of South America, with simulated volumetric moist content ranging from 0.35 to 0.45 $\mathrm{m^3m^{-3}}$ for all soil layers (Figure 5). The high soil moisture of the northwestern Amazonia contrasts with the rest of the region dryness, where the moisture in the top soil layer of the model is below 0.2 $\mathrm{m^3m^{-3}}$ (Figure 5.a), and only deeper soil layers remain fairly moist ( 0.3 $\mathrm{m^3m^{-3}}$ , below 4 m) (Figure 5.c). Comparing soil moisture in two forest areas with different rainfall regimes (red rectangle, 148 mm in average) and blue rectangle (34 mm in average), the area receiving a higher volume of rainfall is about 55% wetter than the other in the shallow layers, but only 12% in the deeper soil layer. By contrast, comparing areas of forest (blue rectangle)



and *cerrado* (gray rectangle) with similar rainfall regime (34 mm), the forest region remains considerably wetter ( 15% both in shallow and deeper layers). According to Köchy & Wilson (2000), high rates of water uptake per unit mass may reflect the high root of the vegetation. In fact, James et al. (2003) found at a site 20 km east of Regina (58°28'N, 104°22'W), Canada, that the ability of grass to reduce soil moisture is nearly five times higher than that of woody vegetation, expressed on a per-gram basis.

For the forest region, the soil moisture values from the model are consistent with the mean value of 0.39 $m^3 m^{-3}$ measured at 0.2 m at *Tapajós* site (near *Santarém*, location indicated in Figure 2) during the dry season (Doughty et al., 2010). Previous measurements at the same site, reported soil moisture only slightly higher at the same site ( 0.44 $m^3 m^{-3}$ , from 0.15 to 0.30 m), and also confirmed the model results for a higher moisture of the deeper soil layers ( 0.42 $m^3 m^{-3}$ at 4 m) (Bruno et al., 2006).

The period chosen for this study coincides with the peak of the biomass burning season when a total of 439,297 fires were detected in the area of the model domain by AVHRR-NOAA, which is 42% of the total number of fires detected by the same sensor during the entire burning season. The spatial distribution of the fires resembles the pattern of precipitation and soil moisture simulated by the model (Figure 5), nonetheless, most of the fires were ignited by human activities (Nepstad et al., 1999; Cochrane & Laurance, 2008). Fires occurred majorly in the *Cerrado*, C3, and C4 type grass-covered areas, but forest

fires, with much higher biomass density, are typically responsible for the highest amount of smoke aerosols and trace gases released into the atmosphere. Figure 6 depicts the spatial distribution of vegetation fires detected by remote sensing over the Amazon and Central Brazil during September 2010.

**Smoke regional plume, CO and CO₂ mixing ratios:** Model results for CO mixing ratio as simulated by the experiment

named DIR+DIF, which it is expected to be the most realistic simulation, were compared with CO measurements made with vertical profile observations made at four different sites in Amazonia during 2010 and 2011 mentioned earlier on. The time series of CO mixing ratio, derived from observation, and model results around 2 km above ground level in four different locations in Amazonia (*Alta Floresta*, *Santarém*, *Rio Branco*, and *Tabatinga*) are shown in Figure 7 (top - left). There was an enhancement of CO from July to October for both years in all the four locations. In 2010, the CO mixing ratio values in

the PBL during the dry season increased from 100 - 150 ppb, typical wet season values (Andreae et al., 2012), to up to 500 ppb, both in the simulations and observations. The dry season of 2010 had CO mixing ratios about twice higher than the same period in 2011, which is consistent with the total number of fires detected by remote sensing during the dry season of these two years which approximately doubled (http://www.inpe.br/queimadas/estatisticas.php). Scatterplots of the CO mixing ratio values from observation and model results for the same four locations, separated into several vertical layers, are depicted in

Figure 7 (top – right). The CO background mixing ratio values for the *Tabatinga* site are close to the 1:1 line, while the biomass burning affected values are more scattered. Model results tend to underestimate observation, especially in lower levels, in the locations mainly affected by fire emissions both locally (*Alta Floresta*, *Rio Branco* and *Santarém*) and by long range transport (*Tabatinga*). This pattern is probably related both to the 20-km model resolution not picking up individual smoke plumes and fire emissions underestimation (Pereira et al., 2016). Previous studies indicated that biomass burning emissions contribute more





than 95% to the variability of CO over the Amazon and that the emissions used in this study (3BEM, Longo et al., 2010) are about 20% underestimated (Andreae et al., 2012).

On the other hand, the airborne vertical profiles analyzed in this study and our modeling results indicated a lesser enhancement of $CO_2$ related to fire activity compared to CO. These results are in agreement with previous measurements of fire smoke

plumes that showed relatively small enhancements of $CO_2$ relative to the background in Amazonia and indicated that fire emissions were not expected to contribute only minorly to $CO_2$ mixing ratios in Amazonia (Andreae et al., 2012). Figure 7 (bottom – left) shows the observed –airborne air sampling at around 2 km above ground level –and simulated time series of $CO_2$ mixing ratio. The biosphere-atmosphere $CO_2$ fluxes respond to a complex myriad of physical processes. Small errors in the configuration of the surface related to the low resolution of the model, can lead to much larger errors in the values of $CO_2$

near the surface. For example, the misplacement of convective systems of few grid cells, very acceptable for a low-resolution atmospheric model, can produce huge variations in the $CO_2$ values near the surface. Still, in general, our model results for $CO_2$ mixing ratios are in an acceptable range of values in comparison with observation. Nonetheless, the $CO_2$ scatter plots (Figure 7, bottom - right) evidenced a much higher variability of both observation values and model results compared to CO, as well as a poorer model representation values close to the ground compared to the upper levels. The low predictability of the

$CO_2$ mixing ratio highlights the complexity of the myriad of processes controlling $CO_2$ fluxes between the surface and the atmosphere and the considerable challenge of modeling them properly. The low-level behavior is likely to be associated with local convective processes but could also have a minor contribution from fresh smoke plumes, both venting $CO_2$ and changing locally the diffuse fraction of solar radiation. In addition, the model struggling to simulate $CO_2$ fluxes could also be related to inaccuracies and low spatial resolution of the soil carbon map (Large Scale Biosphere-Atmosphere Experiment in Amazonia;

Batjes, 1996). Sensitivity studies (not shown here) indicate that in JULES model soil respiration, and consequently $NEE$, are quite sensitive to the prescribed soil carbon content. By contrast, the model tends to better represent the upper levels in terms of observed $CO_2$, which is due to the fact that air circulation is more intense and mainly controlled by the Carbon Tracker boundary conditions, and fire emissions contribution becomes even less significant.

**Smoke regional plume – AOD:** In Figure 8, we show the mean regional smoke plume for September 2010 through the monthly mean of AOD at 550 nm wavelength both from MODIS-Aqua retrieval (Figure 8.a) and from the DIR+DIF model simulation (Figure 8.b). A substantial portion of the Central Brazil and neighboring countries and South of the Amazon Basin were covered by smoke with a resulting monthly mean AOD higher than 0.5, which is 3 to 5 times larger than the typical values of clean conditions. Moreover, also there were large sub-areas with monthly mean AOD higher than 1, indicating a persistent

and high loading of smoke particles. The model results fairly reproduced the spatial distribution of the regional smoke plume from MODIS retrieval. A scatter plot of AOD values from the model and MODIS retrieval (not shown here) had a slope of 0.71 (with $R^2 = 0.73$). Conversely, MODIS retrievals tend to overestimate AOD in relation to the AERONET retrievals in Amazonia, especially for high aerosol loadings (Hoelzemann et al., 2009). Analysis of model results versus AERONET retrieval in some sites in the southern Amazonia (not shown), confirmed that the order of magnitude of the model underestimation is about

the same 20% previously estimated.





**Site level –plant atmosphere CO₂ exchange:** Figure 9.a shows the mean diurnal cycle of $CO_2$ mixing ratio in the first model layer of the three experiments together with the mean diurnal cycle of $CO_2$ mixing ratio just above the canopy of the *Tapajós* forest (near *Santarém*, location indicated in Figure 2) from measurements during September 2010. In *Tapajós*, both observation and model results present a nighttime increase of $CO_2$ due to plant respiration, peaking shortly after sunrise, and a daytime decrease due to photosynthetic processes, with the lowest values before sunset. Despite the model difficulties to simulate the $CO_2$ mixing ratio near surface, in average, the discrepancies with observation were only about 0.9% and 1.4% for the maximum and minimum values, respectively, with the model values lower than the observation during the peak hour and vice-versa when the photosynthetic process dominates. The model amplitude of the $CO_2$ cycle is lower than the observational, though the model cycle is still within the standard deviation of the mean observational diurnal cycle. Model results without including aerosol effects (NO-AER) and with the inclusion of the direct aerosol effect only (DIR-AER) produce a very similar $CO_2$ diurnal cycle. However, the inclusion of the diffuse radiation effects due to biomass burning aerosols reduces the values of $CO_2$ mixing ratio and brings model results much closer to observation, especially during the day, even though the mean AOD modeled in *Tapajós* was very low compared to the area mostly affected by smoke and even underestimated by the model. Considering the whole LBAR area, which includes the region with the highest aerosol load, the inclusion of the aerosol effect on $CO_2$, especially the effect of the diffuse radiation, reduced the $CO_2$ mixing ratio of about 10 ppmv in the $CO_2$ mixing ratio all day long (Figure 9.b). Additionally, in the LBAR the inclusion of the aerosol effects, delays the $CO_2$ diurnal cycle onset in the model results, with the $CO_2$ mixing ratio peaking about 1 hour later. Therefore, the shift of the diurnal cycle of $CO_2$ mixing ratio from the model relative to the observation in *Tapajós* is likely to be related to the AOD underestimation.

In the next section, we will present the model results for energy and carbon fluxes and explore the role played by the smoke aerosol in the carbon cycle in Amazonia.

### 3.2 Impacts of smoke aerosol on energy and carbon fluxes

**Incoming Radiation:** The modeled mean downwelling shortwave irradiance at surface (RSHORT) at 1600 UTC during September 2010 from DIR-AER experiment ranged from 900 $\text{Wm}^{-2}$, in the Southwestern Amazon, to 1000 $\text{Wm}^{-2}$ in the northeastern portion (Figure 10.a), with the smoke aerosol direct impact ($\Delta RSHORT = RSHORT_{DIR-AER} - RSHORT_{NO-AER}$) reaching -100 $\text{Wm}^{-2}$ in smoky areas (AOD > 0.5) (Figure 10.b). As a consequence of the smoke aerosol direct effect, the 2-meter temperature decreased by 1.2 oC in average in the smoky areas around midday ($\Delta Temp = Temp_{DIR-AER} - Temp_{NO-AER}$, Figure 10.c). The noise in the north-western region for both RSHORT and temperature differences within the two simulations is related to expected nonlinear aerosol perturbations on cloud distribution. These results are consistent with previous modeling studies (Rosário et al., 2013) and with estimations based on AERONET measurements (Procópio et al., 2004). As well, observations in *Tapajós* during the dry season indicate an average reduction of 80 and 123 $\text{Wm}^{-2}$, for $AOD > 0.5$ and $AOD > 0.7$, which corresponded to a decrease of the mean temperature of 0.26 and 0.41°C, respectively.





The presence of smoke aerosol in the atmosphere also impacts the flux of PAR radiation in Amazonia during the dry season. Monthly mean PAR radiation ($\mu$mol m$^{-2}$ s$^{-1}$) at 1600 UTC, which is around midday in most of Amazonia, as simulated by the DIR+DIF model experiment for September 2010 is depicted in Figure 11.a. The modeled PAR radiation monthly mean values at 1600 UTC ranged between 900 - 1000 $\mu$mol m$^{-2}$ s$^{-1}$ from southwest to northeast in the LBAR. The presence of

smoke aerosol increases the diffuse fraction of radiation by up to 40% in the smoky areas (Figure 11.b). Figure 12.a shows the diffuse PAR radiation as a function of AOD.

Decreases in surface temperature due to the direct effect of aerosol are also influenced by the balance between latent and sensible heat fluxes, or ultimately, on soil moisture. The difference in 2-meter temperature ($\Delta Temp$) and shortwave irradiance ($\Delta RSHORT$) is, as expected, highly correlated, though with a large band of $\Delta Temp$ for the same value of $\Delta RSHORT$

(Figure 12.b). The $\Delta Temp$ bandwidth increases almost linearly with the $\Delta RSHORT$, with the values correspondent to higher soil moisture populating the lower part of the curve (Figure 12.b), meaning that for regions with the same AOD, the ones with drier soil will suffer higher surface cooling.

**Carbon fluxes for the vegetation types in Amazonia:** Spatial fields for simulated $GPP$ across the northern part of South

America are presented in Figure 13.a for September 2010 at 1600 UTC for forest, C3G, C4G, and shrub in rows 1 -4, respectively. Over forest, simulated $GPP$ ranges from 20 to 25 $\mu$molC m$^{-2}$ s$^{-1}$; while over the regions occupied by *cerrado*, pastures, and tinges of forest there was a much higher variability, with $GPP$ widely varying from below 5 to above 20 $\mu$molC m$^{-2}$ s$^{-1}$. Columns b and c of Figure 13 show the relative impact of aerosol effects on modeled $GPP$ for the 4 biome types (forest, C3G, C4G and shrub, from line 1 to 4, respectively) considering the changes both on direct and diffuse radiation

- in column b, and only on direct solar radiation - in column c.

In Figure 14.a, one can note that the mean solar irradiance that reaches the Amazon region during September promotes a high $GPP$ for the C4 plants, but not high enough to compromise their photosynthesis process. Thus, typically, the $GPP$ of the C4 type plants are highly correlated with the amount of irradiance received, therefore their $GPP$ resembles the same diurnal cycle shape of the irradiance. By contrast, the other vegetation types suffer, in a less or more extent, a decrease in the

carbon assimilation during the period of maximum irradiance, therefore reshaping their $GPP$ diurnal cycle. Still in Figure 14.a, the net increase in $GPP$ due to smoke aerosol over forest ($\Delta GPP_{tot}$, difference between the curves in red and green filled squares) is 3.8 $\mu$molC m$^{-2}$ s$^{-1}$, with the majority ($\%Flux_{diff} \cong 94\%$) of the impact related to the increase in the diffuse fraction of solar radiation. The reduction of direct solar radiation by smoke aerosols increases forest $GPP$ only up to 0.2 $\mu$molC m$^{-2}$ s$^{-1}$ ($\Delta GPP_{dir}$), which is associated with the cooling of the leaves. Over *cerrado* areas, the increase on

$GPP$ was up to 0.9 and 0.1 $\mu$molC m$^{-2}$ s$^{-1}$ due to the aerosol effect on the diffuse fraction radiation ($\Delta GPP_{diff}$), and the direct radiation ($\Delta GPP_{dir}$), respectively, with the aerosol direct radiative effect much lower than the diffuse radiation effect because the $GPP$ of the *cerrado* is also severely limited by the excess of irradiance. In the case of the C4 grass type, which was not limited by irradiance, the direct radiative aerosol effect induced a reduction of the $GPP$ (-0.7 $\mu$molC m$^{-2}$ s$^{-1}$), but the increase of the diffuse fraction of radiation more than compensate the reduction of $GPP$ due the irradiance attenuation of

the direct effect, and, when including both direct and diffuse radiation effects, the $GPP$ jumps from 43 $\mu$molC m$^{-2}$ s$^{-1}$ to 47





$\mu$molC m$^{-2}$ s$^{-1}$. Table 2 resumes the integrated values of $GPP$ for each biome in the LBAR during September 2010, as well as the variation related to the total aerosol effect (both on diffuse radiation and direct radiation, $\Delta GPP_{tot}$), and only with the direct aerosol effect ($\Delta GPP_{dir}$). According to the model results, the net $GPP$ of the forest biome in the LBAR was of 1,206 Tg C during September 2010. The presence of smoke aerosol was responsible for an increase of about 24% of the $GPP$ over

the forest, mainly associated with the impact of the aerosol on the diffuse radiation. For the *cerrado* and C3 grass, the net $GPP$ were 359 and 850 Tg C in the same region, during the same period, with the smoke aerosol acting to increase the $GPP$ of about 16% and 23%, respectively. Rap et al. (2015) using JULES model forced with aerosol fields from another model, estimated an annual increase of the net primary production ($NPP$) ranging between 1.4 and 2.8% related to the aerosol effect in Amazonia. Translating our results to ($NPP = GPP - R_P$), we estimate an increase of 13% for September 2010, 22% related to $GPP$

minus 9% with Rp. Now, considering that the biomass burning season, when the smoke AOD rises above the background values, typically last for about 3 months, we estimate an annual increase of the $NPP$ of about 3.25% related to the aerosol effect. Our results for the aerosol impact on $NPP$ over the Amazonia is slightly higher than the Rap et al. (2015) estimation. However, one must keep in mind that Rap et al. estimation was based on 9 years (1998 – 2007), while our work was based only on 2010, which was a relatively drier and smokier year. We could say then that the two estimations are comparable and within

the annual variability of fire activity. For the C4 grass, the net $GPP$ of 2,431 Tg C increased by only 9% due total aerosol effect, as the direct aerosol effect acted on the contrary reducing the $GPP$. The diurnal cycle of plant respiration for each biome is shown in Figure 14.b for the three model runs. As expected, the higher $GPP$ leads to higher plant respiration. Plant respiration peaks 7.5, 2, 4.6, and 14 $\mu$molC m$^{-2}$ s$^{-1}$ for forest, *cerrado*, C3, and C4 grass types, respectively, with the aerosol impact more pronounced for forest and C4 biomes. The mean soil respiration found in the LBAR is 2.78 $\mu$molC m$^{-2}$ s$^{-1}$, with a relatively

mild diurnal cycle that basically depends on soil temperature, hence with lower values in the morning and a tendency to increase slightly in the afternoon. However, soil respiration is highly variable in the LBAR, depending on the amount of carbon in the soil, being as low as 0.13 $\mu$molC m$^{-2}$ s$^{-1}$ in the *cerrado* and grass-plot areas, and ranging between 2 and 8 $\mu$molC m$^{-2}$ s$^{-1}$ in forest areas. However, we must keep in mind that the soil respiration is strongly dependent on the soil carbon, and a high variability around the Amazon basis is expected. Regarding $NEE$, model results show that the forest biome releases around

+5 $\mu$molC m$^{-2}$ s$^{-1}$ during the night and early morning, and then, when $GPP$ compensates the respiration, the net uptake goes as low as -11 $\mu$molC m$^{-2}$ s$^{-1}$. The net effect is an uptake of approximately 0.015, 0.565, and 0.060 molC m$^{-2}$ day$^{-1}$ for the C3, C4 grass, and forest biomes, respectively, and a release of 0.126 molC m$^{-2}$ day$^{-1}$ for the *cerrado*. Grace et al., (1995), with measurements of CO$_2$ fluxes at Jaru Reserve, Rondônia, Brazil (10°,4.84'S, 61°, 56.60'W) in September 1992, estimated an accumulation of 0.09 molC m$^{-2}$ day$^{-1}$ for forest during the dry season. So, the forest uptake estimation based

on our modeling results is approximately 30% lower than the estimations based on Grace et al. (1995) measurements. However, several measurements in the Amazon indicated both high yearly and regional variabilities around the Amazon Basin due to several factors, which includes hydric stress, aerosol loads, topography, differences in soil carbon and forest physiology. Also, previous studies also indicated that there are high uncertainties in the magnitude of nocturnal $NEE$ measurements during calm nighttime (Araujo et al., 2002; Araujo et al., 2010). In Table 3 we collected from the literature some estimations of $NEE$ (daily

total, nighttime and daytime peak), based on CO$_2$ fluxes measurements in different sites in Amazonia during the dry season

(c) Author(s) 2017. CC-BY 3.0 License.



in different years. For the same sites, we also presented in Table 3, the $NEE$ from DIR+DIF experiment for September 2010. As example, measurements taken at the Jaru reserve (same site used by Grace et al., 1995) and at a grass-plot (Fazenda Nossa Senhora - FNS, at 10°45'44"S, 62°21'27"W), during the dry season from 1999 to 2002, revealed an uptake of around 0.12 and 0.069 $\mathrm{molC\ m^{-2}\ day^{-1}}$ for the pasture and forest site, respectively, with the diurnal values of $NEE$ reaching -13.2 and

-17.5 $\mathrm{\mu molC\ m^{-2}\ s^{-1}}$ for the pasture and forest, respectively (von Randow, 2004). And, more recently, Cirino et al. (2014) reported 10 years of $CO_2$ fluxes measurements carried out in central Amazon's Cuieiras Biological Reserve – K34 LBA (Large Scale Biosphere-Atmosphere Experiment in Amazonia) tower flux (2°36'33" S, 6°12'33" W) from 1999 to 2009, and also in the Jaru reserve from 1999 to 2002. The measured diurnal cycle of $NEE$ in both sites was, as expected, positive during the night time and negative during the daytime. During the dry season, the mean values of $NEE$ measured during the night were

approximately +5.2(±0.8) and +6.8 (±1.0) $\mathrm{\mu molC\ m^{-2}\ s^{-1}}$ at the Jaru (Von Randow et al., 2004) and Cuieras (Cirino et al., 2015) reserves, respectively, due to differences in the physiology of the forest, and maybe topography in the two sites. There were even more significant differences between the maximum values of carbon absorption between the two sites, which were around -17.5 and -20 $\mathrm{\mu molC\ m^{-2}\ s^{-1}}$ under smoky and cloudy sky condition, at the Jaru and Cuieras reserve, respectively, and -18 $\mathrm{\mu molC\ m^{-2}\ s^{-1}}$, under clean-sky in both sites. The differences between the peak values of carbon absorption are

likely related to the presence of smoke, and variability of cloudiness and rainfall. Measurements at the *Tapajós* National Forest (2°51'S,54°58'W) from 2002 to 2005 (during September) led to estimations of $NEE$ varying from +0.017 to –0,069 $\mathrm{molC\ m^{-2}\ day^{-1}}$ (Hutyra et al., 2007). The mean value of $NEE$ from the model at Tapajos was -0.032 $\mathrm{molC\ m^{-2}\ day^{-1}}$ for September 2010. The Jaru reserve is systematically, intensely affected by both local and long range transported smoke aerosols, with monthly means for AOD (550 nm) during the dry season typically above 0.5, but often above 1.0; in contrast,

the smoke affects the northern part of the Basin, where Cuieras and *Tapajós* are located, more episodically than systematically (Longo et al., 2009). In addition, the northern part has more variability in terms of rainfall during the dry season compared to the southern part, due mainly to the position of the Inter-Tropical Convergence Zone (ITCZ). So, a strong variability in the terms of carbon fluxes in the northern part of the Amazon is indeed expected and is really challenging for a low-resolution model to match punctual measurements. Nevertheless, our modeled monthly mean diurnal cycles of $NEE$ for forest (Figure

14.c) are remarkably close to the diurnal cycle reported for the Jaru reserve and FNS by von Randow et al., (2004), with the total daily total, nighttime, and daytime peak values (Table 3) of similar order of magnitude within the variability observed. Figure 15 depicts the model results of DIR+DIF simulation for $GPP$ ($\mathrm{\mu molC\ m^{-2}\ s^{-1}}$) in response to the available PAR ($\mathrm{\mu mol\ m^{-2}\ s^{-1}}$) for the main four different biome types in the domain of study and in a range of 0.2 to 0.6 of the fraction of the diffuse irradiance. The analysis is also limited for the condition where the soil wetness is above 0.9. The maximum values

of $GPP$ for Forest and C3 type grass are reached with PAR around 1,600 $\mathrm{mol\ m^{-2}\ s^{-1}}$ and 1,300 $\mathrm{mol\ m^{-2}\ s^{-1}}$, respectively, indicating that these are the saturation point for these biomes relative to the amount of energy reaching the surface. For *cerrado* (shrub), the saturation point is much lower, around 600 $\mathrm{mol\ m^{-2}\ s^{-1}}$, but the plants maintain its carbon assimilation rate up to PAR around 1,600 $\mathrm{mol\ m^{-2}\ s^{-1}}$, only then decreasing with further increase of PAR. For the C4 type grass, $GPP$ increases almost linearly with PAR, not showing any evidence of saturation with the amount of energy received. In general, the results



in Figure 15 indicate that, for a given biome and amount of PAR, higher fraction of the diffuse radiation implies higher $GPP$. However, the *cerrado* (shrub) biome is an exception since it saturates with relatively lower values of PAR.

The mean spatial distribution of the relative impact of aerosol effects on modeled fluxes at 1600 UTC is shown in Figure 16 considering the changes both on direct and diffuse radiation, and only on direct solar radiation. In the LBAR region, where

there is predominance of the forest biome but not only, there is an increase of $GPP$ (Figure 16.b.1) ranging from 0.1 to 5.0 µmolC m$^{-2}$ s$^{-1}$, related to the aerosol effect, the lower values being associated with lower AOD values (Figure 8) and drier soil (Figure 5). In the remaining regions, the aerosol impact on $GPP$ is still positive but much lower ($0.1 - 0.5$ µmolC m$^{-2}$ s$^{-1}$). In all domain, the majority of the aerosol impact is related to the increase in the diffuse fraction of solar radiation due to the presence of aerosols ($\%GPP_{diff} > 95\%$). As a general rule, our model results indicated that the increase of $GPP$ leads to

an increase ranging between 0.2 to 1.6 µmolC m$^{-2}$ s$^{-1}$ on plant respiration ($R_P$) for the forest and does not affect the other biomes significantly. On the other hand, soil respiration ($R_H$, line 3 of Figure 16) varies from 1 to 8 µmolC m$^{-2}$ s$^{-1}$, with the higher values in the forest area with higher soil moisture (Figure 5). The aerosol impact on $R_H$ is somewhat noisy, varying from -1 to +1 µmolC m$^{-2}$ s$^{-1}$ over the forest, negative otherwise ( -0.1 µmolC m$^{-2}$ s$^{-1}$). The noise of $R_H$ is associated with the non-linear effects of aerosol on cloudiness and precipitation. The total impact of aerosol in $NEE$ (line 4 of Figure 16)

ranged from -0.1 to -5 µmolC m$^{-2}$ s$^{-1}$, with the lower values found in the forest region with intense/persistent smoke, mainly related to the diffuse radiation effect, meaning that the smoke aerosol effect creates a CO$_2$ sink in Amazonia.

**Total carbon fluxes in the Amazonia, weighting for the vegetation types:** Figure 17 depicts the monthly mean diurnal cycle of the CO$_2$ fluxes in the LBAR for September 2010, again related to $GPP$, $R_P$, $R_H$, and $NEE$, averaged over the

4 types of vegetation present in each atmospheric model grid box. The presence of the smoke aerosol affects the CO$_2$ flux associated with $GPP$, and consequently Rp. Responding to the increasing diffuse radiation, both $GPP$ and $R_P$ rise, being the $GPP$ enhancement about 4 times higher than the $R_P$. On contrast, RS has an opposite response as the presence of aerosol implies in a cooler soil (Figure 10) and, consequently, lower bacterial activity. The net effect is a higher CO$_2$ daytime uptake, with a negligible night-time variation. Moreover, the smoke aerosol strongly impacts the $NEE$ (Figure 17.b). Around noon,

the $NEE$ decreases from -7 to -10 µmolC m$^{-2}$ s$^{-1}$ in the presence of smoke, mainly due to the diffuse radiation effect. Nevertheless, it is interesting to notice that the relative contribution of the diffuse to the total (diffuse + direct) aerosol effect on the $NEE$ (Equation 4) has a quite distinct behavior depending on the biome type and exponentially decay, or increase, with the AOD increase for all biomes, and C4 grass type, respectively. The contribution of the diffuse radiation effect to $NEE$ ($NEE_{\%diff}$ calculated according to Equation 6) is depicted in Figure 18. The fitting functions for the $NEE_{\%diff}$ versus

AOD for each biome are on the plot. Over forest, the percentage of the diffuse radiation effect on CO$_2$ uptake decreases exponentially ($[NEE_{\%diff}]_{forest} \approx e^{-0.9AOD}, R^2 = 0.7$) from 100% to 50% with the increase of aerosol loading, reaching a balance of 50% - 50% between the diffuse and direct effect, narrowing the spreading, with AOD above 0.5. For C3 grass type and *cerrado*, as expected, the contribution of the diffuse radiation effects tends to near zero with the increase of AOD ($[NEE_{\%diff}]_{cerrado,C3} \approx 0.7e^{-4AOD}, R^2 = 0.7$). While for C4 grass type, the contribution of the diffuse radiation to $NEE$

exponentially increases with AOD ($[NEE_{\%diff}]_{C4} \approx e^{AOD}, R^2 = 0.9$), as this biome does not saturate with the amount of





energy received. Considering the AOD underestimation of about 20% and the exponential behavior of the relative contribution of the diffuse fraction to $NEE$, it is fair to say that the contribution of the diffuse radiation effect on $CO_2$ uptake can reach 40% over the forest, and 10% over *cerrado* and C3 grass type, for high aerosol loads.

The model results for the $CO_2$ fluxes integrated for the month of September (2010), which is the peak of the burning season in the LBAR, are summarized in Table 4. The $GPP$ in the LBAR is 1,113 Tg C, with the aerosol being responsible for an increase of 240 Tg C, with less than 1% due to the aerosol radiation direct effect. The plant respiration is affected by approximately 50 Tg C, related only to the increase of the diffuse fraction of radiation. The impact of the aerosol on the soil respiration is only 3% but in the opposite direction. Integrating throughout the full month for September 2010, the $NEE$ changed from +101 Tg C to -104 Tg C, when the aerosol effect is considered. The total aerosol effect on radiation was responsible for about 96% of the $NEE$ change, while the temperature reduction due to the direct aerosol effect on radiation accounts for only 5%. That is, the aerosol effect, especially the change in the diffuse fraction of radiation, is strong enough to invert the signal of $NEE$, changing the ecosystem from being a source to a sink of $CO_2$. The difference between modeling and observational estimation for $NEE$ is likely to be within the yearly, and spatial variability of forest ecosystem physiology, which also includes disturbed areas and secondary forest.

## 4   Final remarks

We conducted a modeling study during the peak of the burning season in Amazonia to assess the ability of a current state-of-the-art integrated in-line numerical atmospheric modeling system to simulate the $CO_2$ fluxes in Amazonia. A set of three different modeling experiments, first totally disregarding biomass burning aerosol effect, then considering only the direct aerosol effect, and, finally, also adding the aerosol effect on the diffuse fraction of radiation. The model results allowed us to assess and quantify the impacts of smoke aerosols on $CO_2$ fluxes in the Amazon Basin during the dry season. Moreover, the relative role of the main soil/vegetation and atmosphere interaction processes controlling the carbon cycle in Amazonia was weighed and the aerosol effect on each of them was measured separately.

Consistently with previous studies (Freitas et al., 2005, 2009, and 2016; Longo et al., 2010, 2013; Rosário et al., 2013; Moreira et al., 2013), BRAMS performed well while modeling the meteorology and smoke aerosol emission, transport and removal processes in Amazonia, which has resulted in fairly simulation of the major features of AOD variability associated with the regional smoke plume over South America. The model results for surface temperature, rainfall and AOD were once again in agreement with observations for the 2010 dry season case study, representing the main characteristics of the spatial distribution and the diurnal cycle of temperature and precipitation. BRAMS was also evaluated on its performance to simulate CO and $CO_2$ mixing ratios using results acquired from measurements on samples airborne collected over the Amazon during 2010 and 2011 burning seasons. Typically, the model tends to slightly underestimate the CO mixing ratio, particularly in the lower levels, in regions affected by fresh smoke and haze smoke layers. Previous studies had already indicated an underestimation of the biomass burning emissions database used in this work (3BEM, Longo et al., 2010) of about 20% (Andreae et al., 2012), mainly related to fire omission and misrepresentation of the vegetation and carbon maps used (Pereira et al., 2016). For $CO_2$





mixing ratios, the comparison between model and observation is highly scattered, again specially in the lower levels, though in this case more likely related to convective activity pumping $CO_2$ to the upper layers of the atmosphere. In both cases, model inaccuracies are to be, at least partially, related to the lower model resolution (20 km), and then further sensitivity studies on model resolution are highly recommended. Nevertheless, although the 20-km model resolution was not capable of capturing

punctual $CO_2$ measurements in the Amazonia, the order of magnitude of the $CO_2$ mixing ratio has been in general well represented. Moreover, the diurnal cycle of $CO_2$ measured on top of the canopy of the *Tapajós* forest was represented in the model with differences of only about -0.9% and +1.4% between model results and observations during the time of minimum and maximum values, respectively.

Our modeling results indicate that during the dry season in Amazonia, regions with lower precipitation not always have high

values of $NEE$ because the lower soil respiration of a dryer soil can compensate the deficit of water available for plants. Being an equatorial region, Amazonia abundantly receives PAR radiation, then areas with plenty water availability in the soil have higher $GPP$ compared to dry soil areas. However, around noon local time, when the energy excess typically occurs, there is a drop in carbon assimilation for all biomes, except for the C4 grass type that has a maximum assimilation coinciding with the peak of PAR radiation.

The presence of an intense smoke aerosol layer during the dry season over the Amazonia reduces the solar energy reaching the surface, and consequently reducing near surface temperature. The model results show this cooling effect contributing to increasing the $GPP$ in regions covered by forest, grass C3 and *cerrado*. However, in addition to reducing the surface energy, the aerosol layer also increases the diffuse fraction of radiation, and this is the major effect that contributes to increasing the $GPP$, and, in this case, including the C4 grass type biome. These two effects all together, increase the $GPP$ of about 25%,

22%, 8% and 16% for forest, C3, and C4 type grasses, and *cerrado*, respectively.

In the LBAR, the $GPP$ increased about 22%, reaching 1,113 TgC during September 2010, when the aerosol effect was included. The plant respiration also increased from 510 to 560 TgC, with the smoke aerosol effect as a response to the increase of $GPP$. The more $CO_2$ the plant assimilates to produce sugar, more it needs to increase its respiration for energy supply. On the other side, soil respiration dropped from 463 to 449 Tg C. Consequently, the $NEE$ in the LBAR during September 2010

dropped from +101 to -104 TgC when the aerosol effects were considered, mainly due to the diffuse radiation effect. That is, the LBAR during the dry season, in the presence of high smoke aerosol loads, change from being a source to be a sink of $CO_2$ to the atmosphere.

These results are also consistent with Yamasoe et al. (2006) observations, who had found no correlation between $NEE$ and aerosol load for low AOD values (< 0.7); however, for AOD > 0.7 $NEE$ values became negative, and for AOD > 1.5-2 $NEE$

started to increase again. Our model results also indicated that the impact of the aerosol on the $NEE$ change is mainly related to the aerosol increasing the diffuse fraction of radiation.

For AOD higher than 0.5, the forest reaches a balance of 50% – 50% between the diffuse and direct aerosol effects. For C3 grass type and *cerrado*, as expected, the contribution of the diffuse radiation effect is much lower than for the forest biome and tends to near zero with the increase of AOD. Direct measurements at the *Tapajós* site (Doughty et al., 2010) led to an estimation

of the relative aerosol contribution in $CO_2$ uptake, for high values of AOD, of 80% as a result of increased shaded light in the





sub-canopy, related to the effect of aerosol increasing the diffuse fraction of radiation. While only 20% of the aerosol impact on $CO_2$ uptake was attributed to the decreased of canopy temperature. These same authors, however, do recognize that is "difficult to know whether this proportion is applicable to forest biomes worldwide or limited to tropical forest". So, based on our model results, we go even further and say that it is difficult even to affirm that there is a unique rule applicable to the all Amazon forest due to its high diversity, heterogeneity, and microclimates.

Considering that the fire activity in Amazonia typically last for about 4 months, from June to October, we can estimate that, due to the aerosol-radiation interaction, the LBAR absorbs 416 TgC instead of releasing 404 TgC to the atmosphere. The net impact of the smoke aerosols on the carbon cycle in Amazonia is about -820 TgC per year. According to Espírito-Santo et al. (2014), the impact of the natural disturbance in the carbon cycle in Amazonia is approximately 1,300 TgC per year. Thus, the aerosol (negative) impact is of a similar order of magnitude of the (positive) impact of the natural disturbances in the carbon cycle in Amazonia.

Our model results lead us to highly emphasize the importance of considering the effects of aerosol in numerical models of climate forecasting, especially when investigating the intensification of the greenhouse effect due to the atmospheric $CO_2$ concentration. In general, the numerical results obtained were in a solid agreement with observational data, including meteorological, aerosol and trace gases variables, which gives us a high degree of confidence in the estimates of the carbon fluxes. However, we do recognize that further model development based on current level of knowledge could still improve the representation of biomass burning aerosol effects in the carbon cycle. As such, model studies that include the reduction of photosynthesis due to the oxidation of plant leaves by high levels of ozone secondarily produced in biomass burning plumes, as well as the indirect aerosol effect on the $CO_2$ are undergoing. As well, we will soon report the inclusion of the cloud effect on the increasing of the diffuse fraction of solar radiation in the model, which is certainly a major effect on the $CO_2$ budget in Amazonia during the wet season.

*Author contributions.* D. S. Moreira and K. M. Longo prepared the manuscript. L. M. Mercado, S. R. Freitas, M. A. Yamasoe, E. Gloor, and N. E. Rosário reviewed the manuscript. D. S. Moreira, S. R. Freitas, K. M. Longo, N. E. Rosário contributed on BRAMS code development. L. M. Mercado contributed on JULES code development and on JULES-BRAMS models coupling. D. S. Moreira designed the numerical experiments and carried them out. D. S. Moreira, K. M. Longo, S. R. Freitas, and R. S. M. Viana worked on model validation/evaluation and model results analysis. M. A. Yamasoe and N. E. Rosário provided aerosol and radiation observational data and analysis. Finally, J. B. Miller, L. V. Gatti, K. T. Wiedemann, L. K. G. Domingues and C. C. S. Correia provided carbon fluxes observational data.





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





**Table 1.** Parameters for a three-degree polynomial fit to the diffuse fraction of broadband solar irradiance reaching the surface as function of AOD at 670 nm for distinct air mass intervals.

| Optical air mass | a | b | c | d | $R^2$ |
|---|---|---|---|---|---|
| m ≤ 1.10 | 0.0115 | -0.1115 | 0.4693 | 0.1258 | 0.994 |
| 1.10 < m ≤ 1.25 | 0.0129 | -0.1235 | 0.4997 | 0.1304 | 0.994 |
| 1.25 < m ≤ 1.40 | 0.0075 | -0.1087 | 0.5035 | 0.1477 | 0.989 |
| 1.40 < m ≤ 1.70 | 0.0052 | -0.1031 | 0.5077 | 0.1795 | 0.990 |
| 1.70 < m ≤ 2.00 | 0.0144 | -0.1634 | 0.6207 | 0.1696 | 0.991 |
| 2.00 < m ≤ 2.80 | 0.0166 | -0.2237 | 0.7458 | 0.1851 | 0.981 |
| m > 2.80 | 0.0736 | -0.4631 | 1.0152 | 0.2005 | 0.985 |



**Table 2.** Monthly mean values of net $GPP$, $\Delta GPP_{tot}$ and $\Delta GPP_{dir}$ (all in $\mathrm{TgC\,month^{-1}}$) in the LBAR during September 2010 for the three simulations and different biomes.

| Biome | $GPP_{DIR+DIF}$ | $\Delta GPP_{tot}$ | $\Delta GPP_{dir}$ |
|---|---|---|---|
| Forest | 1,206 | 293 | 8 |
| C3G | 850 | 195 | 24 |
| C4G | 2,431 | 200 | -69 |
| *Cerrado* | 359 | 59 | 12 |





**Table 3.** $NEE$ measurements during the dry season in several locations in the Amazon region, Brazil. The mean values and standard deviation of $NEE$ from the modeling results from DIR+DIF experiment (September 2010) are also presented in the last column.

| Loca | Biome type | Data collection period | NEE from flux measurements | | | NEE from the model (Sept 2010) | | |
|---|---|---|---|---|---|---|---|---|
| | | | Daily total $molCm^{-2}day^{-1}$ | Nighttime $\mu molCm^{-2}s^{-1}$ | Daytime peak $\mu molCm^{-2}s^{-1}$ | Daily total $molCm^{-2}day^{-1}$ | Nighttime $\mu molCm^{-2}s^{-1}$ | Daytime peak $\mu molCm^{-2}s^{-1}$ |
| Jaru[a] | | Sept 1992 | -0.090 | - | - | -0.065 ±0.002 | +5.2 ±0.8 | -11.9 ±2.4 |
| Jaru[b1] | Forest | dry season 1999-2002 | -0.069 | +7.1 | -17.5 | | | |
| FNS[b2] | Pasture | Dry season 1999-2002 | -0.12 | +3.0 | -13.2 | -0.14* ±0.1 | +3.3 ±0.8 | -13.1 ±5.4 |
| *Tapajós*[c] | Forest | Sept 2002 | +0.017 | - | - | -0.032 ±0.039 | +5.6 ±0.9 | -11.9 ±0.6 |
| | | Sept 2003 | + 0.026 | - | - | | | |
| | | Sept 2004 | -0.017 | - | - | | | |
| | | Sept 2005 | -0.069 | - | - | | | |
| Sinop[d] | Forest | Dry season 2005–2006 | +0.008 ±0.029 | +5.2 ±0.4 | - | -0.23 ±0.006 | +3.0 ±0.3 | -14.4 ±1.0 |
| | | Dry season 2006–2007 | -0.013 ±0.024 | +5.5 ±0.4 | - | | | |
| | | Dry season 2007–2008 | -0.041 ±0.022 | +5.6 ±0.3 | - | | | |
| Cuieiras[e] | Forest | Dry season 1999-2009 | - | ∼+4.0 | -20.0 | +0.037 ±0.015 | +6.8 ±1.0 | -11.7 ±1.4 |

[a] Jaru reserve (10°05'S, 61°57'W), Grace et al., 1995.

[b1] Jaru reserve (10°05'S, 61°57'W), Von Randow et al., 2004.

[b2] Fazenda Nossa Senhora (10°45'44"S,62°21'27"W), Von Randow et al., 2004.

[c] *Tapajós* National Forest (2°51'S,54°58'W), Hutyra et al., 2007.

[d] 50 km NE of Sinop-MT (11°25'S, 55°20'W), Vourlitis et al., 2011.

[e] Cuieiras Biological Reserve (2°36'33" S, 60°12'33" W), Cirino et al., 2015.

[*] The pasture site was identified as C4 grass type.

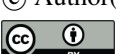



**Table 4.** Total $CO_2$ assimilation ratio for the DIR+DIF simulation, $\Delta Flux_{tot}$ and $\Delta Flux_{dir}$ (all in $\mathrm{TgCmonth^{-1}}$) in the LBAR during September 2010 for the different atmosphere-biosphere exchange processes.

| Process | $CO_2\ Flux$ | $\Delta Flux_{tot}$ | $\Delta Flux_{dir}$ |
|---------|--------------|---------------------|---------------------|
| $GPP$   | 1,113        | 240                 | 1                   |
| $R_p$   | 560          | 50                  | 0                   |
| $R_H$   | 449          | -14                 | -7                  |
| $NEE$   | -104         | -205                | -8                  |





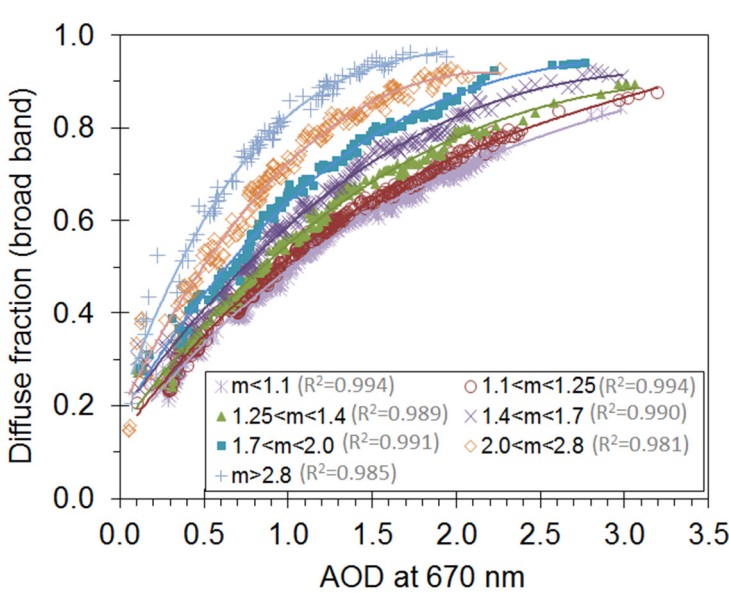

**Figure 1.** Fraction of diffuse broadband solar irradiance reaching the surface as function of AOD at 670 nm and optical air mass intervals (m).



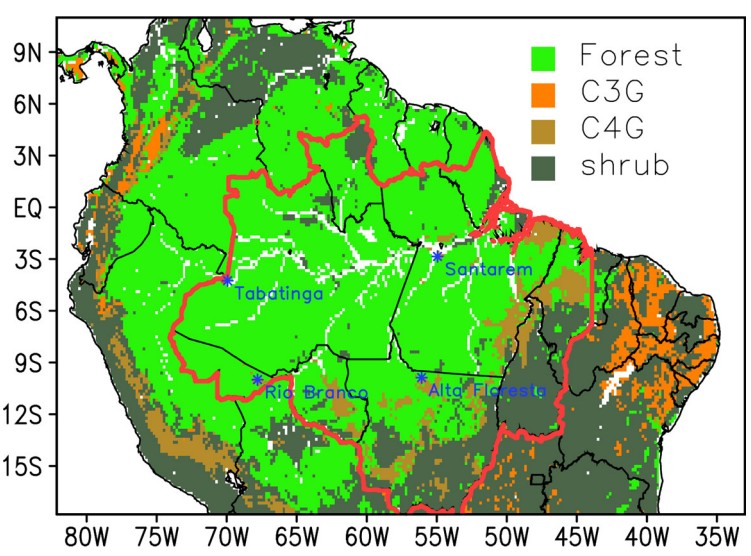

**Figure 2.** Model domain with the map of land cover used in BRAMS, the color scale depicting the dominant biomes. The red contour line on the map delimits the LBAR and the locations of $CO_2$ and CO airborne measurements: *Santarém*, PA (2.85° S, 54.95° W); *Rio Branco*, AC (9.99° S, 67.80° W); *Alta Floresta*, MT (9.87° S, 56.09° W); and *Tabatinga*, AM (4.25° S, 69.94° W).





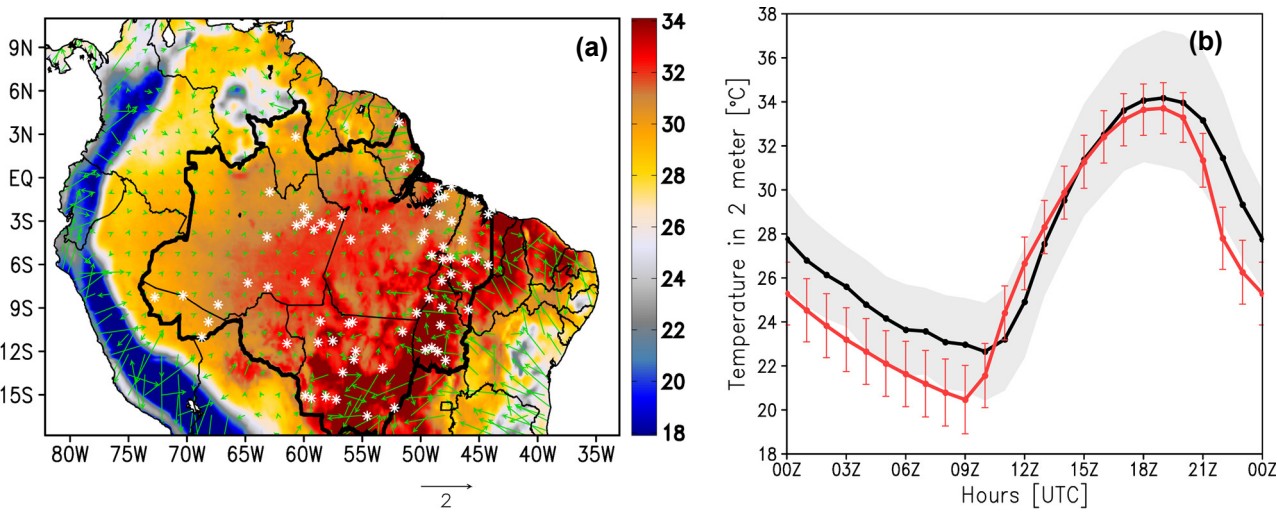

**Figure 3.** (a) Spatial distribution of mean 2-meter temperature and 10-meter wind field from the model during September 2010 at 1600 UTC. The LBAR and the locations of 72 near-surface measurement ground stations, used to evaluate the model 2-meter temperature, are depicted in the map with white asterisks. (b) Mean diurnal cycle of 2-meter temperature (°C) observed in the 72 near-surface ground stations (black line) and from the model grid-point nearest the respective station (red line). The standard deviation of mean temperature from observation and from the model results are indicated by shaded gray and red bars, respectively.



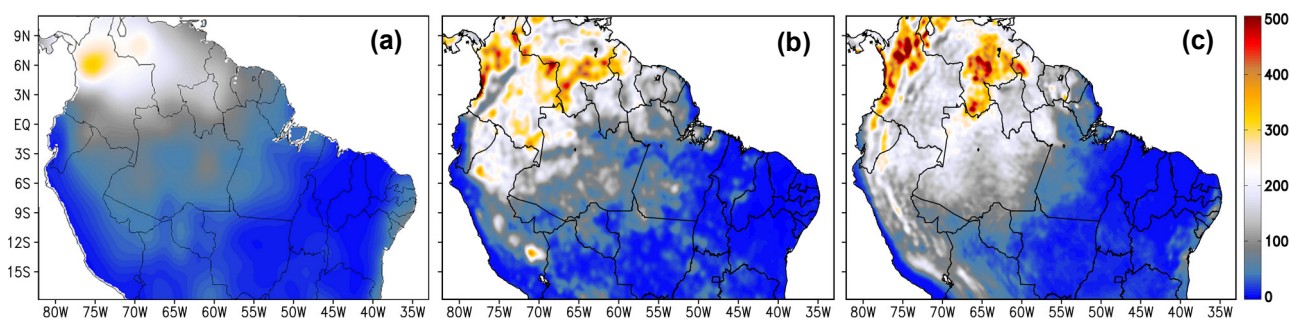

**Figure 4.** Accumulated precipitation (mm) during September 2010 from the (a) ground stations observation network interpolated for the model grid point, (b) TRMM rainfall product, and (c) model results.




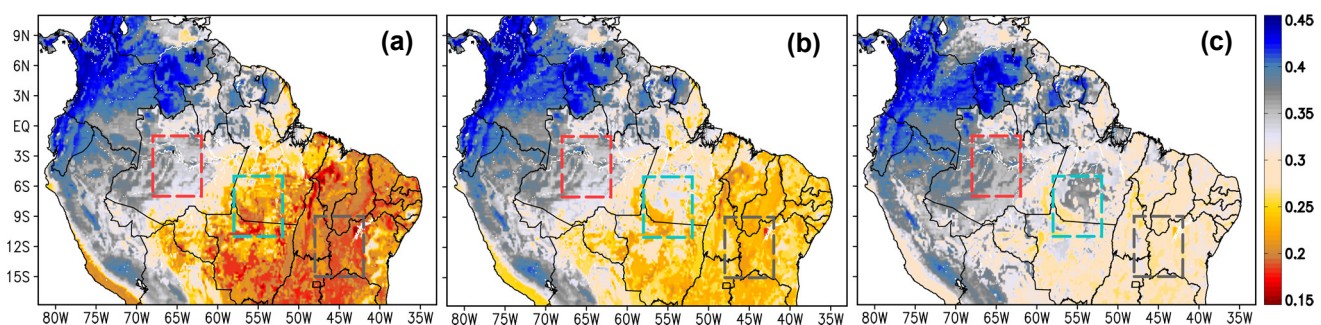

**Figure 5.** Monthly mean soil moisture ($\mathrm{m^3 m^{-3}}$) during September 2010 at three soil levels: (a) 0.35 m, (b) 1.00 m, and (c) 4.25 m. The rectangles depict areas with predominance of forest and moist soil (red), forest and dry soil (blue), and *cerrado* with dry soil (gray).



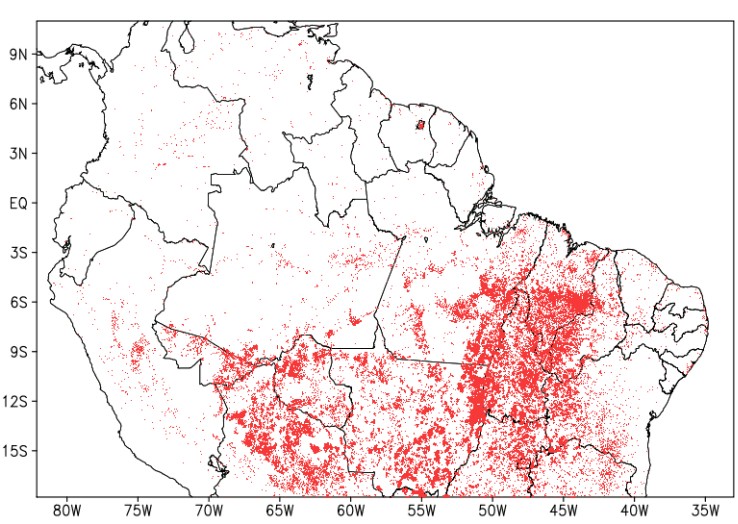

**Figure 6.** Fire product derived from AVHRR measurements during September 2010. (Source: www.cptec.inpe.br/queimadas).



**Figure 7.** Time series (on the left) and scatter plots (on the right) of CO (ppbv) (on the top) and $CO_2$ (ppbv) (on the bottom) airborne measured (black dots) and simulated (DIR+DIF experiment, blue dots and line) at *Alta Floresta*, *Santarém*, *Rio Branco*, and *Tabatinga* (indicated on each plot), at about 2 km above the ground level from April 2010 to October 2011. On the time series, the red dashed lines on the time series indicate the fire season of 2010. On the scatter plots, the colors scale depicts the vertical layers: < 900 m (purple), 900-1800 m (blue), 1800-2700 (green), 2700-3600 (orange), and > 3600 m (pink), and the error bars refer to the standard deviation of the mean values. The locations of the measurement sites are indicated in Figure 2.



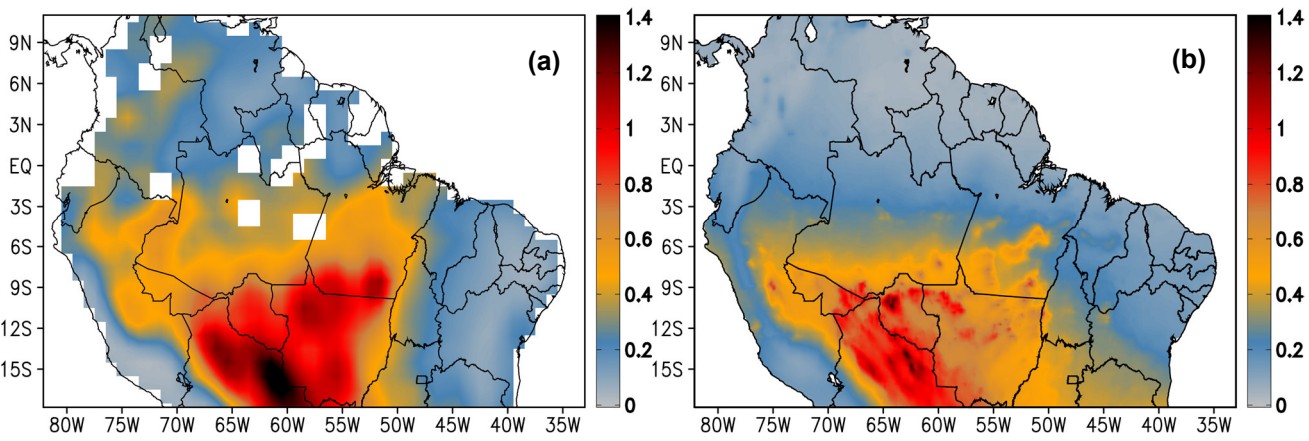

**Figure 8.** Monthly mean AOD at 550 nm wavelength for September 2010 from (a) MODIS Aqua retrieval, and (b) from the model as simulated on the DIR+DIF experiment.





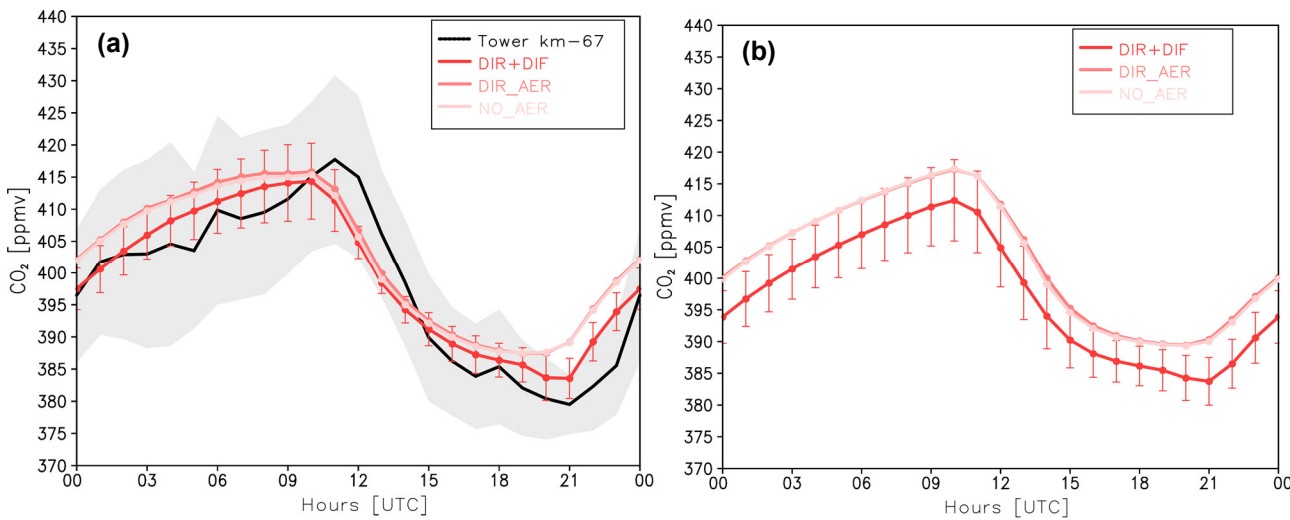

**Figure 9.** (a) Mean diurnal cycle of $CO_2$ (ppmv) mixing ratio (black line) during September 2010 in *Tapajós* forest tower at 39.6 meter
(2.85° S, 55.04° W, slightly southward *Santarém*, location indicated in Figure 2). The gray shaded area indicates the standard deviation of
mean values observed. (b) Mean diurnal cycle of $CO_2$ (ppmv) mixing ratio from the model during the same period in the LBAR (red line in
Figure 2). In both plots, model results are at the 39.3 meters' model level for the three simulations NO-AER (light pink), DIR-AER (pink),
and DIR+DIF (red). The red error bars on the model curves refer to the standard deviation of the mean model values.

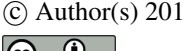



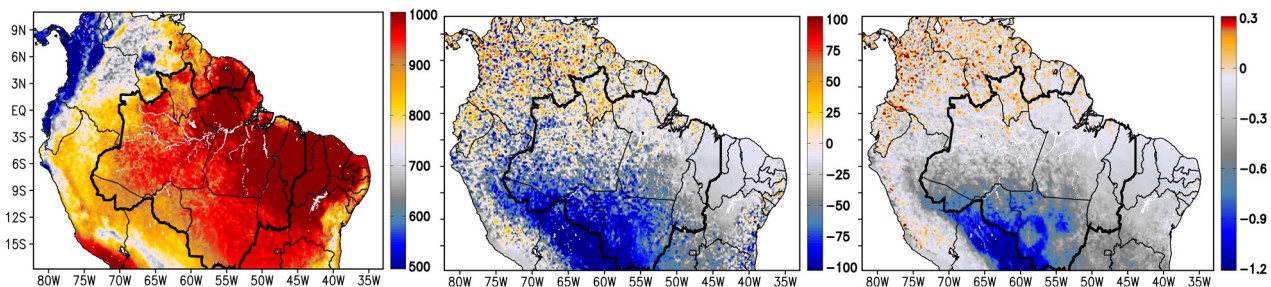

**Figure 10.** (a) Mean downwelling shortwave irradiance at the surface ($\mathrm{Wm}^{-2}$) at 1600 UTC (which is around midday in most of Amazonia) for September 2010 from DIR-AER experiment; (b) the difference in the mean downwelling shortwave irradiance at the surface ($\mathrm{Wm}^{-2}$) during the same time period as simulated at DIR-AER and NO-AER; and (c) the difference in the 2-meter temperature (°C) during the same time period as simulated at DIR-AER and NO-AER. The darker black contour line on the maps delimits the LBAR.





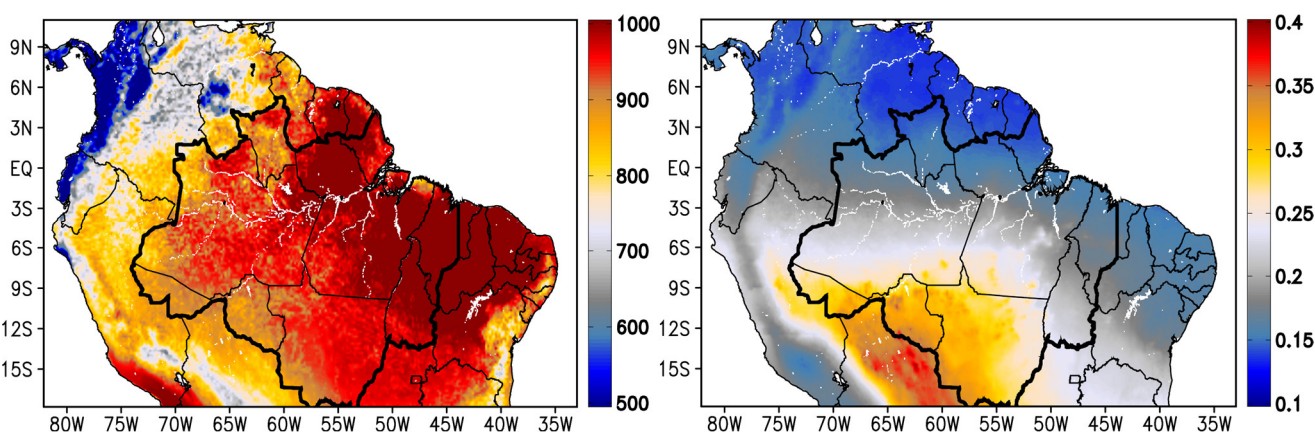

**Figure 11.** (a) Mean PAR radiation ($\mu$mol m$^{-2}$s$^{-1}$) at 1600 UTC for September 2010 from DIR+DIF experiment; and (b) the mean diffuse fraction of solar radiation at 1600 UTC for September 2010 as simulated by DIR+DIF. The darker black contour line on the maps delimits the LBAR.





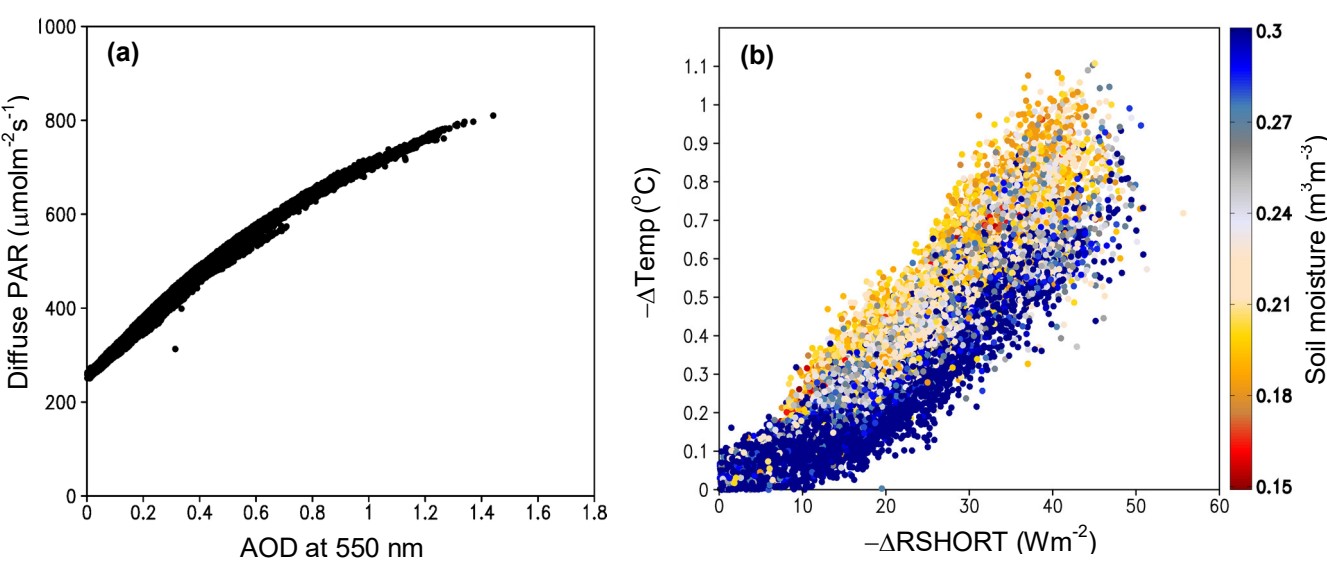

**Figure 12.** (a) The diffuse PAR radiance ($\mu$molm$^{-2}$s$^{-1}$) versus AOD at 550 nm as simulated at DIR+DIF experiment at 1600 UTC during September 2010 in the LBAR. (b) The decrease of 2-meter temperature versus the decrease of downwelling shortwave irradiance (Wm$^{-2}$) as simulated at AER-DIR and NO-AER experiments at 1600 UTC during September 2010 in the LBAR. The color scale refers to soil moisture (m$^3$m$^{-3}$) at 0.35 meter.





**Figure 13.** Mean $GPP$ (μmolCm$^{-2}$s$^{-1}$) for September 2010 at 1600 UTC as simulated at DIR+DIF (column a), the difference in the monthly mean $GPP$ (μmolCm$^{-2}$s$^{-1}$) as simulated at DIR+DIF and NO-AER (column b), and DIR-AER and NO-AER (column 6). The lines from the 1st to the 4th present the results for the biomes of forest, C3G, C4G, and shrub, respectively. The darker black contour line on the maps delimits the LBAR.





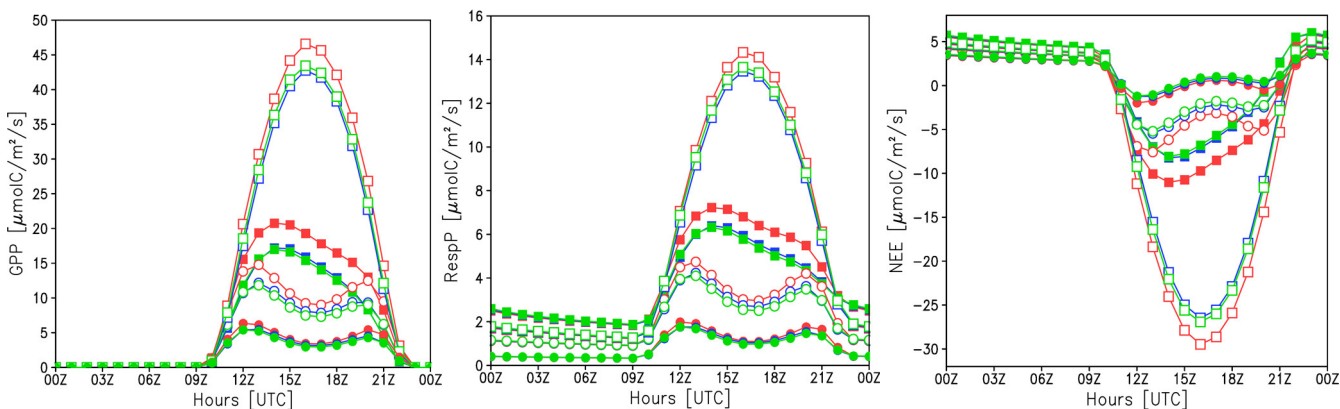

**Figure 14.** Mean diurnal cycle of (a) $GPP$ (µmolCm$^{-2}$s$^{-1}$), (b) $R_P$ (µmolCm$^{-2}$s$^{-1}$), and (c) $NEE$ (µmolCm$^{-2}$s$^{-1}$) during September 2010 for the different biomes in the LBAR. Different symbols indicating the biomes of forest (filled squares), C3G (hollow circles), C4G (hollow squares), and shrub (filled circles); and different colors indicating the modeling experiments, green for NO-AER, blue for DIR-AER and red for DIR+DIF.





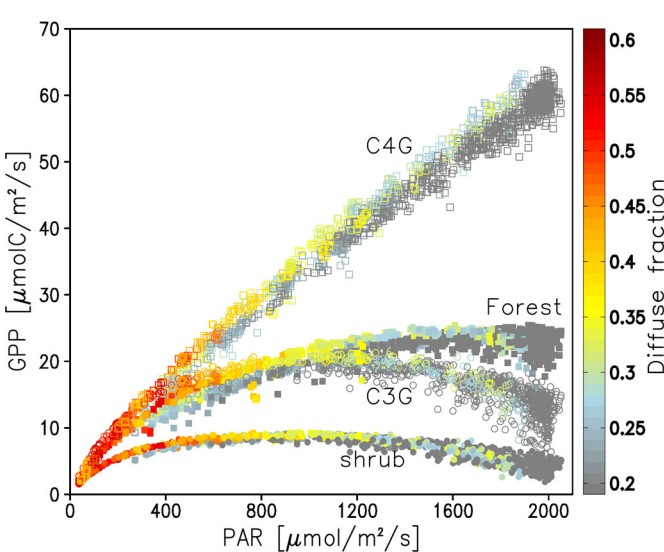

**Figure 15.** Figure 15: Scatter plots of $GPP$ (μmolCm$^{-2}$s$^{-1}$) and PAR radiation (μmolm$^{-2}$s$^{-1}$) from the DIR+DIFF experiment for forest (filled squares), shrub (filled circles) biomes, grass types C4 (hollow squares) and C3 (hollow circles) during September 2010 in the LBAR. The color scale depicted indicates the fraction of diffuse radiation. The data were filtered for soil water factor above 0.9.



**Figure 16.** Mean net $CO_2$ fluxes ($\mu mol\,Cm^{-2}s^{-1}$), weighted per biome type, for September 2010 at 1600 UTC (column a), and the differences in the mean $CO_2$ fluxes as simulated at DIR+DIF and NO-AER (column b), and DIR-AER and NO-AER (column c) during the same time period. The lines from the 1st to the 4th have the results for $GPP$, $R_P$, $R_H$, and $NEE$ processes, respectively. The darker black contour line on the maps delimits the LBAR.





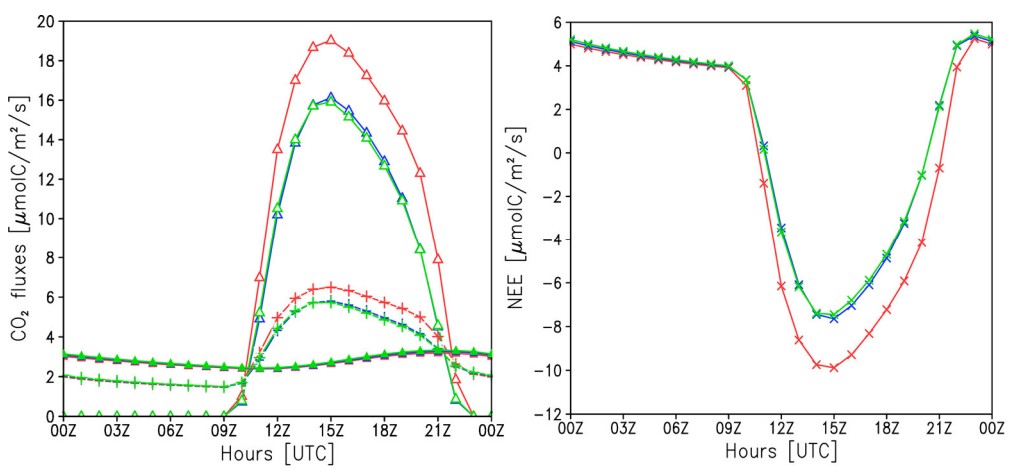

**Figure 17.** (a) Mean diurnal cycle of $CO_2$ fluxes ($\mu$molCm$^{-2}$s$^{-1}$) during September 2010 in the LBAR, with different symbols indicating the processes of $GPP$ (hollow triangle), $R_P$ (plus) and $R_H$ (filled triangles). (b) Mean diurnal cycle of $CO_2$ fluxes ($\mu$molCm$^{-2}$s$^{-1}$) associated with $NEE$ during the same period and in the same area. In both plots, different colors indicating the modeling experiments, green for NO-AER, blue for DIR-AER and red for DIR+DIF.





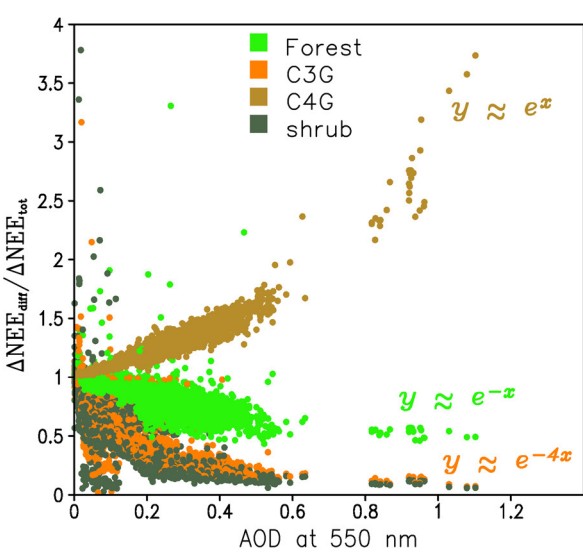

**Figure 18.** The contribution of the diffuse radiation effect to $NEE$ ($NEE_{\%diff}$ calculated according Equation 6) as a function of AOD in the LBAR, but separated with different colors for different types of vegetation. The model data were filtered for cloudiness and precipitation. Additionally, only model points with the same soil water factor within all the three experiments, and soil moisture difference below 0.001 $m^3 m^{-3}$ were included. The fitting functions of the $NEE_{\%diff}$ versus AOD for each biome are also shown in the figure.