# Peer review of "Modelling the radiative effects of biomass burning aerosols on carbon fluxes in the Amazon region"

_Atmospheric Chemistry and Physics, 2016_

## Referee Comment (RC1) · Anonymous Referee #2 · 18 Mar 2017

This study uses a modelling approach to quantify the impact of biomass burning aerosol on CO2 fluxes through changes in direct and diffuse surface solar radiation in the Amazon region. Assessing and improving the ability of atmospheric models to simulate such effects is important and this study can potentially contribute to this effort, therefore being in principle well-suited for ACP. The manuscript is reasonably well written and easy to follow in its logic. I have several (mainly) minor recommendations that I would like to see addressed before publication.

Specific comments:

- p 4, l 15: not clear what you mean by the two-way mode coupling, could you please describe in more detail how this coupling is implemented and how it works

- p 5, l 1-3: did I understand correctly that all other aerosol emissions, except biomass

burning, were ignored in the model? If this is the case, I would like the authors to add a few words here on why this assumption is needed from a technical point of view and what inaccuracies is likely to introduce (e.g. neglecting masking effects and interactions from other aerosol types etc.). I think that rather than doing a "no aerosol" vs. "biomass burning only" comparison, it would be preferable to do a "all aerosol" vs. "no biomass burning" comparison.

- p 5, l 28: please explain in a bit more detail how the cloud filtering was done. Also need to say how ignoring the effect from clouds is likely to affect results presented in this study, preferably also attempting the quantify this.

- p 7, l 8-9: I struggle to understand why you did not run all 3 simulations for the whole 2-year period (as you did for the DIR+DIF experiment) and I would strongly recommend to do so. The ability to make annual estimations would substantially increase the significance of the paper.

- p 10, l 11-12: Fig 3b shows in fact that the model values are outside the standard deviation range of the observed temperatures for all night and late afternoon hours

- p 11, l 31: here and in other parts throughout the paper where you compare modelled vs. observed values, please quantify these comparisons by giving some relevant stats (e.g. mean bias, correlation etc.)

- p 12, l 20: can you add a reference to these sensitivity studies?

- p 12, l 31,34: why are these results not shown as they seem to be important here?

- p 13, l 18: "CO2 mixing ratio peaking about 1 hour later" – this is actually not apparent from Fig 9b

- p 14, l 18-20: here you should discuss in more detail what these columns b-c actually show and what it means. Also, can you evaluate the results presented in Fig 13 against some observed values?

[Figure]

- p 15, l 4: here and throughout the manuscript, please revise the way you calculated all percentage changes. If GPP increased by 293 Tg C month-1, from 913 to 1206, this means an increase of 32%, not 24%.

- p 15, l 9-10: I don't quite understand how you derived the 13% increase in NPP. If A=B-C and B increases by 22% and C increases by 9%, this does not imply that A increases by 13%. Please clarify.

- p 15, l 10-12: I don't think you can make such extrapolations (one peak season month is by no means representative of the entire season). Also, here you say that the biomass burning season lasts for 3 months (and thus divide by a factor of 4), while later (p 20, l 6) you say that it lasts 4 months (and use that for other estimates). These comparisons really needs to be addressed properly and in addition to correcting the current mistakes, it is very apparent that the paper would benefit a lot from performing annual simulations for all 3 experiments.

- p 19, l 28-30: can you include a direct comparison of your model results with these Yamasoe et al conclusions, i.e. do you also see the same behaviour for these AOD intervals?

- p 39, Fig 9a: should explain in the text why is the effect (difference between the red and pink line) stronger during the night?

In addition, I think the readability of the paper could be substantially improved by getting editing help from someone with full professional proficiency in English.

Technical corrections:

- title: I suggest a slight change of title, replacing "Amazon" with "Amazonia" or "the Amazon region"

- use the present tense in the abstract when presenting your results, e.g. "our results indicate..." etc

- p 2, l 4: "to be a sink" -> "to being a sink"

- p 2, l 8: "cerrado" -> "cerrado areas"

- p 2, l 14: "areas of about several" -> "areas of several"

- p 2, l 14: "out of the biomass burning season" -> "outside the biomass burning season"

- p 2, l 18: "Angstrom exponent" -> "The Angstrom exponent"

- p 2, l 23-26: please rephrase, possibly removing the first phrase which is unnecessary

- p 3, l 6: "deplete" -> "achieve"

- p 3, l 10: not clear what you mean by "net radiation"; do you mean "total radiation"?

- p 5, l 23: define D (from eq 1) somewhere in the text

- p 7, l 21: ":" -> "." (or rephrase using small letter after the colon, as it implies that a list of things is following)

- p 8, l 6-7: not clear if you want ratios or percentages here

- p 8, l 21 & p11, l 22: "observation" -> "observations"

- p 9, l 35: best to use consistently throughout the manuscript either "biomass burning" or "smoke"

- p 10, l 12: please rephrase "diurnal cycle early in 1-hour"

- p 13, l 27: "oC" -> degree C

- p 14, l 1: here and throughout the manuscript, PAR already includes "radiation", so no need to say "PAR radiation"

- p 14, l 35: "GPP jumps" – please rephrase, e.g. "GPP increases"

- p 15, l 1: "Table 2 resumes" –> "Table 2 summarises"

- p 16, l 24: "punctual" –> "point"

- p 17, l 1: do you mean "lower fraction of diffuse radiation"?

- p 17, l 18: "weighting" –> "weighted"

- p 17, l 25-28: not clear what you mean here, please rephrase

- p 18, l 2: please replace "it is fair to say" with a more scientific wording

- p 18, l 12-14: not sure what you mean here, please rephrase this last sentence of the paragraph

- p 19, l 19: remove "all" from "all together"

- p 27, Table 1 caption: "three-degree" –> "third-degree"

- p 33, Fig 3b caption: please clarify what standard deviations are shown for the black and red lines (e.g. what values were used to derive them)

- p 34, Fig 4: since this shows precipitation, I suggest to reverse the colour scale (as you did for Fig 5)

- p 36, Fig 6 caption: please clarify what exactly you mean by "fire product"

- p 37, Fig 7: please include some stats (here or in the text) for the scatter plots on the right. Also, the standard deviations are missing from the top scatter plots. I would also suggest to use a better colour scale for altitude to help visualising the results (at the moment the purple and pink are too similar – a more intuitive transition from low to high altitudes is preferable)

- p 39, Fig 9: please use different colours for the model results (the light pink is almost invisible). Since you use UTC, it might help to show with dashed vertical lines where the local sunrise/sunset times are on the X-axis.

- p 40, Fig 10: why not showing the effect of the best simulation (DIR+DIF)?

- p 44, Fig 14: please add a legend

- p 45, Fig 15: please clarify what values are shown here (spatially and temporally)

- p 47, Fig 17: please add a legend

———————————————

---

## Referee Comment (RC2) · Anonymous Referee #1 · 21 Mar 2017

The paper describes an interesting modelling exercise and seems to be sufficiently backed up by measurements to warrant the output to be within reasonable limits. The results demonstrate an interesting and to some degree surprising effect of the high aerosol load over the Amazon basin. The topic is obviously suitable for publication in ACP and in my opinion has the potential to attract an interested readership.

Unfortunately, the wording is often quite particular (shouldn't the title read '...in the Amazon region' or similar), despite the English language being overall comprehensible. Examples of such a particular wording which provides wrong spelling, twisted logic as well as unusual usage of words are P5L9 "...mixing ratios are diagnosed from the prognostic variables using the saturation mixing ratio with respect to liquid water", P12L6 "...fire emissions were not expected to contribute only minorly to $CO_2$ mixing ratios...", or P14L22 "... a high GPP for C4 plants, but not high enough to compromise

their photosynthesis process". A considerable language editing should therefore be carried out. This can be accompanied with extensive shortenings, particularly towards the end of the manuscript (i.e. P16-20).

Furthermore, the paper needs more emphasize on the biosphere model. The reader only learns that the JULES model has been used and that it had been evaluated for sites in the Amazon before. What has not been explained in the methodology section is how the model considers direct and diffuse radiation for photosynthesis or how this response depends on plant functional type. It is also important to know how radiation and temperature changes influence simulated respiration (calculating a fixed or variable fraction of photosynthesis being lost as 'growth respiration', exponential temperature dependence on maintenance respiration, allocation shifts regarding exudation or fine root turnover changes the effect decomposition,...?). The depicted model properties (simplifications) should be used in the discussion to point out the appropriateness of the processes or the need for improvements.

One of the reasons why the sensitivity of the model is important is that the importance of the direct and indirect aerosol effects might actually been less important than it looks like. I refer to chapter 3.2 where it is mentioned that the direct aerosol effect (by shading) reaches -100 Wm-2 (Tapajos 80-123 Wm-2), which comes along with a certain amount of cooling. This corresponds to about 460 umol m-2 s-1 global radiation or roughly speaking 230 umol m-2 s- PAR reduction. On the other hand, Fig. 12 shows that the increase of diffuse PAR due to the indirect aerosol effect is from app. 250 to 800 = 550 umol m2 s-1. If direct and diffuse radiation are similarly effective in the model (please explore), the aerosol effect by shading should thus be about half the magnitude of the increase in diffuse radiation. Since it seems to be smaller, the cooling effect (part of direct aerosol effect) seems to compensate for the greater part of the shading. In my opinion this should be discussed in greater detail, using the sensitivity of the model against temperature changes for argumentation.

Some more specific notes:

P7L4: I don't understand what is meant by 'spin up artifacts'. Usually spin-ups are used to avoid artifacts originating from uncertain initial conditions.

P7L7ff: From Fig. 12 it is apparent that diffuse PAR is about 250 umol m2-s-1 under conditions of AOD = 0 (clear sky conditions). I guess that this is about 5 percent of the total radiation even if AOD is actually 0. It seems likely that some clouds are even increasing this fraction. On the other hand the DIR-AER scenario seems to exclude this part of the radiation, which causes a bias that underestimates radiation and thus photosynthesis. Can you comment on this?

P8L6ff: Equations 5 and 6 seem superfluous to me. A short description in the text should suffice.

P8L26/28: Why are there two different algorithm numbers (3B42 and as 3B43)?

P10L22: What is meant by 'several precipitation systems'?

P10L27ff: The description of the soil moisture is a bit confusing. I would like to know how the soil is considered and initialized in the model (soil depth, number of layers, stratification of potential water content).

P11L10: The number of fire needs a reference. It seems to be considerable higher what is given in Chen et al. 2013 (Biogeosciences, Vol. 118, P495ff).

P11L19ff: I don't see any connection between the CO concentration and the biosphere model (but I may be wrong), which would mean that the DIR-AER and DIR+DIF scenarios should result in very similar concentration distributions. Is this correct? The simulation of CO concentrations seems to serve primarily for showing that the physical processes involved are correctly represented in the atmospheric model. This should be highlighted.

P12L8ff: I think it should be clearly articulated that the model fails to represent the CO2 concentrations. Model results are clearly not 'in an acceptable range' for most of the sites and periods. The reasons seem not to be clear but I am sure that some of

the most likely ones can be depicted instead of blaming a 'complex myriad of physical processes'. I would differentiate between uncertainties in transport and biosphere exchange processes. If the authors render air chemical reactions as important despite the relative small reactivity, they might include them too. It should be noted, however, that blaming biosphere process uncertainties (including uncertainties in soil drought determination) means to question GPP and NEE results. Overall, I would suggest clearly arguing that the model is not sensitive to the $CO_2$ concentration within the given range (app. 385-395) and that therefore the model problems should not have a major effect on final results.

P15L14: Here it is firstly indicated that the investigated year might not be representative for the general conditions ('a relatively drier and smokier year'). This should be discussed further. To which degree differs the year from others? Which effect might this have on the overall results?

P15L21-24: I have the impression that for this analysis, it is decisive to evaluate the difference of respiration (and other fluxes) between scenarios. The variability in soil conditions that is certainly influencing the absolute magnitude seems to be less important.

P18L15ff: I recommend refraining from an additional summary like it is done with the 'Final remarks' section and instead create a 'conclusions' section that points out what has been learned from the analysis and what should be considered in future research.

Figure 7: Note that observations are generally depicted on the x-axis while simulation results are shown on the y-axis.

Figure 9: I am missing the effect of the scenarios on total/direct radiation.

---

## Author Comment (AC1) · 1 Jul 2017

We thank the referee for his(er) insightful and very helpful comments, which contributed to improve the paper. The answers to his(er) questions and comments are below:

**Legend:**   RC: Referee's Comment
AR: Author's response
AC: Author's changes in manuscript

[Figure]

**Specific comments**

1. **RC: p 4, l 15: not clear what you mean by the two-way mode coupling, could you please describe in more detail how this coupling is implemented and how it works**

   AR: The coupling is considered two-way in the sense that, for each model time step, the atmospheric component provides to JULES the current near-surface wind speed, air temperature, pressure, condensed water and downward radiation fluxes, as well as water vapor and trace gas (e.g carbon dioxide and monoxide, methane, and volatile organic compounds) mixing ratios. After its processing, JULES advances its state variables over the time step and feeds back the atmospheric component with the sensible and latent heat and momentum surface fluxes, upward shortwave and longwave radiation fluxes, and a set of trace gas fluxes . Further details on the JULES x BRAMS coupling is described in Moreira et al., 2013, nevertheless we included a sentence about it in the current manuscript.

   AC: ...has been coupled in a two-way mode with the Joint UK Land Environment Simulator v3.0 (JULES), the land surface scheme of the UK Hadley Centre Earth System model, as described in Moreira et al. (2013). The coupling is two-way in the sense that, for each model time step, the atmospheric component provides to JULES the current near-surface wind speed, air temperature, pressure, condensed water and downward radiation fluxes, as well as water vapor and carbon dioxide mixing ratios. After its processing, JULES advances its state variables over the time step and feeds back the atmospheric component with sensible and latent heat and momentum surface fluxes, upward shortwave and longwave radiation fluxes, and a set of trace gas fluxes.

2. **RC: p 5, l 1-3: did I understand correctly that all other aerosol emissions, except biomass burning, were ignored in the model? If this is the case,**

**I would like the authors to add a few words here on why this assumption is needed from a technical point of view and what inaccuracies is likely to introduce (e.g. neglecting masking effects and interactions from other aerosol types etc.). I think that rather than doing a "no aerosol" vs. "biomass burning only" comparison, it would be preferable to do a "all aerosol" vs. "no biomass burning" comparison.**

AR: Correct, the model was run with only biomass burning aerosols. We do agree that it would be ideal to do "all aerosol" vs. "no biomass burning". However, natural sources of aerosols (biogenic and soil dust) are not yet functional in the model and we don't have a good inventory for urban emission sources for the Amazon region and northern of Brazil. Because of this issue, the urban emission was turned off. Nevertheless, regarding the aerosol loading across the Amazon basin and neighborhood, in the absence of biomass burning, AOD in the visible spectrum hardly overcomes 0.15, which would translate in a very low radiative impact compared with that observed under massives AOD values (very often above 1.0) that occur during biomass burning influence. Therefore, from the point of view of the radiative effect impacts, we would not expect to see substantial changes in the current results doing "all aerosol" vs "no biomass burning", but, yes, it would be preferable and more consistent.

3. **RC: p 5, l 28: please explain in a bit more detail how the cloud filtering was done. Also need to say how ignoring the effect from clouds is likely to affect results presented in this study, preferably also attempting the quantify this.**

AR: The cloud filter was used only in the interpretation of the model results, i.e. we considered only the model gridboxes where the total column integrated condensed water was equal zero.

AC: However, as discussed below, the analysis presented here focuses only on areas, and during hours, without cloud cover, i.e. the results were obtained by filtering out the points with cloudiness, considering only the model gridboxes where

the total column integrated condensed water was equal to zero. The aim of this work is only to compute the aerosol effect; therefore this filter was essentially used to exclude the effects of the clouds in the $CO_2$ fluxes.

4. **RC: p 7, l 8-9: I struggle to understand why you did not run all 3 simulations for the whole 2-year period (as you did for the DIR+DIF experiment) and I would strongly recommend to do so. The ability to make annual estimations would substantially increase the significance of the paper.**

   AR: As the paper focus is on the evaluation of the biomass burning aerosols radiative effect impacts, which are only relevant during the dry season, we decided to focus on the biomass burning period. We agree that run all the experiment for the whole 2 year would add value to the paper. However, that would be interesting if we were able to evaluate accurately the clouds impact. The model current version does not have a robust and well tested parameterization to compute the diffuse radiation from clouds. As the reviewer may know, in the wet season the atmosphere in the Amazon region is dominated by clouds and it is very clean, as biomass burning aerosols or from any other source are almost absent. Therefore, since we are not able to consider the cloud effect with the current model version, the results of the three simulations would be very similar for the wet season. Therefore, not justifying to run the three for a full year . We are currently working on the inclusion of the diffuse radiation effect from clouds in our model, and we plan to extend the analysis of present study for a broader period in the near future.

5. **RC: p 10, l 11-12: Fig 3b shows in fact that the model values are outside the standard deviation range of the observed temperatures for all night and late afternoon hours**

   AR: That is correct, the model is slightly outside the standard deviation of the observation during the night and the afternoon. We have made the correction in

the article.

AC: Changed "... was typically cooler ( $\sim$2.5 C) during the night period but still within the standard deviation of the mean temperature observed." to "... was typically cooler ( $\sim$2.5 C) during the night period and late afternoon hours, and was not far from the standard deviation of the observed mean temperature"

6. **RC: p 11, l 31: here and in other parts throughout the paper where you compare modelled vs. observed values, please quantify these comparisons by giving some relevant stats (e.g. mean bias, correlation etc.)**

AR: We included, in Figure 7, the parameters of a linear fitting to the scatter plot and the respective R-Squared (see Figure below).

AC: "Model results tend to underestimate $CO$ and $CO_2$ observations, especially at low levels, in locations mainly affected by fire emissions both locally (*Alta Floresta*, *Rio Branco* and *Santarém*) and by long range transport (*Tabatinga*). The black line on each scatter plot in Figure 7 shows the linear fit and the correspondent R-Squared values. The largest $CO$ underestimation occurred in *Alta Floresta*, with a slope of coefficient equal 0.58, but the highest dispersion occurred in *Santarém*, with $R^2 = 0.58$."

7. **RC: p 12, l 20: can you add a reference to these sensitivity studies?**

AR: We included in the supplementary document figures showing the JULES sensitivity to some variables. Please, see Figures S.2, S.3 and S.4 below.

AC: We performed sensitivity tests to assess JULES response to several atmospheric variables. We ran JULES offline (version 3.0) for September 2010 using as input BRAMS results for the NO-AER experiment considering the nearest gridbox to the Tower km-67. Figure S.1 (in the supplementary material) shows monthly variation of shortwave radiation (Rshort), longwave radiation (Rlong), air temperature near surface, specific humidity near surface, all used as input for

the sensitivity test. The soil carbon in this gridcell is 10 $kgCm^{-2}$ and constant during all the month. Besides the BRAMS model results for each parameter, we also varied each parameter, reducing and increasing its original value to cover the standard deviation of the monthly mean. In addition, we varied the diffuse fraction of the shortwave radiation, which was originally zero (NO-AER scenery), from 0 to 0.8 of the total radiation (Rshort). Therefore, we ran 567 simulations for the month of September 2010. For each simulation, we calculated the monthly mean fluxes. Figures S.2, S.3 and S.4 show the results for these sensitivity tests. JULES results for soil respiration, and consequently $NEE$, are quite sensitive to the prescribed soil carbon content (Figure S.2). In addition, the $GPP$ increases with the increase of soil moisture for all biomes (Figure S.3). However, $R_H$ and $R_P$ also increases with the soil moisture (Figure S.3a and S.3m). Therefore, for the forest and *cerrado* biomes, the $NEE$ decreases until a certain value, after then increases again with the increasing of soil moisture (Figure S.3s). In summary, the sensitivity analyses show that i) for a 7% decrease in shortwave radiation there are minimal changes in $GPP$ (Figure S.4a); ii) a change in temperature of one degree Celsius (from current midday conditions) also did not imply in major changes in the simulated $GPP$ (Figure S.4b); and iii) a 40% increase in the diffuse fraction of shortwave radiation increased the $GPP$ by 39% , 71%, 4%, and 72% in forest, C3, C4 grasses, and *cerrado* (shrubs) vegetation , respectively (Fig S.4c).

8. **RC: p 12, l 31,34: why are these results not shown as they seem to be important here?**

    AR: A figure describing the results was included in the supplementary document (see Figure S.5 below). Now we are mentioning this within the text:

    AC: Changed "A scatter plot of AOD values from the model and MODIS retrieval (not shown here) had a slope of 0.71 (with $R^2 = 0.73$)" to "The scatter plot of AOD values from the model and from MODIS retrievals (Figure S.5 in supplementary

material) presents a slope of 0.71 (with $R^2 = 0.73$) "

9. **RC: p 13, l 18: "CO$_2$ mixing ratio peaking about 1 hour later" – this is actually not apparent from Fig 9b**

   AR: You are right, the CO$_2$ peak for the three simulations occurred at 10 UTC. We removed this phrase.

10. **RC: p 14, l 18-20: here you should discuss in more detail what these columns b-c actually show and what it means. Also, can you evaluate the results presented in Fig 13 against some observed values?**

    AR: We included a more detailed description about columns b and c. In addition, we showed the evaluation of model results against observed values in table 3 and Figure 9a. And, as described in the text in lines 23-26 on page 4, the evaluation of simulated diffuse radiation effects on $GPP$ using JULES under primarily high or primarily low diffuse radiation conditions has been done for two flux sites in the Amazon rainforest (*Tapajós* and French Guyana) by Rap et al. (2015) and for temperate ecosystems in Mercado et al. (2009).

    AC: In Column b of the Figure 13 we show the difference between monthly mean $GPP$ as simulated for the DIR+DIF and NO-AER experiments, i.e. the relative impact of the total effect of aerosols on simulated $GPP$ for the 4 studied biome types: forest (b.1), C3G (b.2), C4G (b.3) and *cerrado*(b.4). In column c, we show the difference between monthly mean $GPP$ of the simulation without the aerosol effect on the diffuse radiation (DIR-AER) and the simulation without any aerosol effects (NO-AER), i.e. we evaluate the relative impact on the direct solar radiation effect.

11. **RC: p 15, l 4: here and throughout the manuscript, please revise the way you calculated all percentage changes. If GPP increased by 293 Tg C month-1, from 913 to 1206, this means an increase of 32%, not 24%.**

AR: You are right, there were some miscalculation of the percentages, which has been corrected.

AC: p15, l 4: "...an increase of about 32% of the $GPP$..."

AC: p15, l 7: "... about 20% and 30%, respectively."

AC: p19, l 19-20: "...increase the $GPP$ of about 32%, 30%, 9% and 20% for forest, C3, and C4 type grasses, and *cerrado*, respectively."

12. **RC: p 15, l 9-10: I don't quite understand how you derived the 13% increase in NPP. If A=B-C and B increases by 22% and C increases by 9%, this does not imply that A increases by 13%. Please clarify.**

AR: You are right, the NPP estimate was wrong, we made the correction.

AC: P15L7-10 ... about 20% and 30%, respectively. We estimated an average increase of 27% in $GPP$ for the month of September 2010 in the LBAR region, associated to the aerosol effect in Amazonia (Table 4). However, Rap et al. (2015), using JULES model forced with aerosol field from another model, estimated an average increase in $GPP$ of only 2.8% for August, considering the period of 1998-2007. Also, our estimative of net primary production ($NPP = GPP - R_P$) for the simulation DIR+DIF was 553 TgC/month (1,113 - 560) and for the simulation NO-AER was 363 TgC/month ((1,113-240)-(560-50)). Therefore, we estimated an increase of 52% in $NPP$ for September 2010, due to the presence of biomass burning aerosol in LBAR region, while Rap et al. (2015) estimated an increase in $NPP$ of only 5.4% in August. Our results for the aerosol impact over the Amazonia is higher than the Rap et al. (2015) estimation. However, one must keep in mind that Rap et al. estimation was based on 9 years (1998 – 2007) and for a month (August) that typically has much lower aerosol loading than September, while our work was based on September, the peak for the biomass burning season, and for 2010, a drier and smokier year.

13. **RC: p 15, l 10-12: I don't think you can make such extrapolations (one peak season month is by no means representative of the entire season). Also, here you say that the biomass burning season lasts for 3 months (and thus divide by a factor of 4), while later (p 20, l 6) you say that it lasts 4 months (and use that for other estimates). These comparisons really needs to be addressed properly and in addition to correcting the current mistakes, it is very apparent that the paper would benefit a lot from performing annual simulations for all 3 experiments.**

AR: Yes, in the beginning we estimated the duration of the fire season as 4 months. However, after analyzing the monthly fire count data in Brazil from 1999 to 2016 we realized that 3 months was a better estimation (Figure S.7a). So, we decided to maintain the extrapolation with 3 months and kept it consistent all over the text. However, it is important to point out that 2010 is the second top year in terms of fire detection, only surpassed by 2004, and it is out of the standard deviation of the mean from 1999-2015 (Figure S.7b). So, the extrapolation is valid for the years when the fire activity is most intense but it is not representative of an average year.

We are currently working on a longer term model simulation that will allow us to explore the individual and combined aerosol and cloud effects during a full seasonal cycle results.

14. **RC: p 19, l 28-30: can you include a direct comparison of your model results with these Yamasoe et al. conclusions, i.e. do you also see the same behaviour for these AOD intervals?**

AR: We removed the phrase in the final version of the manuscript. Although the figure below illustrates that our modeling results are similar to Yamasoe et al. (2006) observations for low AOD values, in 2010 maximum AOD values were lower than in 2002 making it difficult to extrapolate the model results for the higher AOD interval. Moreover, observational results present higher variability and can

be attributed to the difficulty in controlling all the variables affecting the estimation, such as wind speed, air temperature and humidity, soil moisture, and in removing respiration from the $CO_2$ fluxes measurements to estimate NEE.

AC: Our model results indicated that the impact of the aerosol on the NEE changes is mainly related to the aerosol increasing the diffuse fraction of radiation, as suggested by Yamasoe et al. (2006).

15. **RC: p 39, Fig 9a: should explain in the text why is the effect (difference between the red and pink line) stronger during the night?**

   AR: This is a really interesting observation. These observations belong to the fetch of a flux tower (20Km x 20km) located near the Tapajos River, with 43% Forest cover, 24% water and 32% C3G coverage. A possible explanation may be related to neighboring influences. Observe in Figure S.6 (below) that the mean wind at 00 UTC is from East (forest region) bringing air mass that has carbon fluxes affected by aerosol during the previous daytime. Meaning that this grid point receives air mass from a region where the effect of the aerosol was more pronounced, leading to a greater difference between the simulation without aerosol and with aerosol.

   AC: A curious fact is that at night the difference between NO-AER and DIR+DIF is greater than in the daytime period. One possible explanation for this is the influence of the neighborhood. Note in Figure S.6 of the supplementary material that the average wind at 00 UTC is from East, a forest region, and has differences between aerosol and non-aerosol simulations. However, the wind at 10 UTC is coming from NorthEast, crossing the river, where the influence of the aerosol in the carbon fluxes is low.

16. **RC: In addition, I think the readability of the paper could be substantially improved by getting editing help from someone with full professional proficiency in English.**

AR: The document has been reviewed by a co-author with English as his first language.

**Technical corrections**

1. **RC: title: I suggest a slight change of title, replacing "Amazon" with "Amazonia" or "the Amazon region"**

   AR: We agree.

   AC: We changed the title to: "Modelling the radiative effects of biomass burning aerosols on carbon fluxes in the Amazon region"

2. **RC: use the present tense in the abstract when presenting your results, e.g. "our results indicate: : :" etc**

   AR: Thank you.

3. **RC: p 2, l 4: "to be a sink" → "to being a sink"**

   AR: Thank you.

4. **RC: p 2, l 8: "*cerrado*" → "*cerrado* areas"**

   AR: Thank you.

5. **RC: p 2, l 14: "areas of about several" → "areas of several"**

   AR: Thank you.

6. **RC: p 2, l 14: "out of the biomass burning season" → "outside the biomass burning season"**

   AR: Thank you.

7. **RC: p 2, l 18: "Angstrom exponent" → "The Angstrom exponent"**

   AR: Thank you.

8. **RC: p 2, l 23-26: please rephrase, possibly removing the first phrase which is unnecessary**

   AR: It was rephrased.

9. **RC: p 3, l 6: "deplete" → "achieve"**

   AR: Thank you.

10. **RC: p 3, l 10: not clear what you mean by "net radiation"; do you mean "total radiation"?**

    AR: You are right, the more adequate term is "total radiation". This has now been changed.

11. **RC: p 5, l 23: define D (from eq 1) somewhere in the text**

    AR: This is now defined.

    AC: "D represents the diffuse fraction and the values of the fitting parameters a, b, c, and d, of..."

12. **RC: p 7, l 21: ":" → "." (or rephrase using small letter after the colon, as it implies that a list of things is following)**

    AR: Changed ":" to "."

13. **RC: p 8, l 6-7: not clear if you want ratios or percentages here**

    AR: The equations 5 and 6 were removed.

14. **RC: p 8, l 21 & p11, l 22: 'observation' → 'observations'**

    AR: Thank you.

Interactive
comment

15. **RC: p 9, l 35: best to use consistently throughout the manuscript either "biomass burning" or "smoke"**

    AR: "smoke" was changed to "biomass burning" in all over the text.

16. **RC: p 10, l 12: please rephrase "diurnal cycle early in 1-hour"**

    AR: It was rephrased.

    AC: "In addition, the model temperature has a diurnal cycle with a gap of one hour more early than the observation."

17. **RC: p 13, l 27: "oC" → degree C**

    AR: Thank you.

18. **RC: p 14, l 1: here and throughout the manuscript, PAR already includes "radiation", so no need to say "PAR radiation"**

    AR: We removed the word "radiation". Thank you.

19. **RC: p 14, l 35: "GPP jumps" – please rephrase, e.g. "GPP increases"**

    AR: We replaced "jumps" by "increases". Thank you.

20. **RC: p 15, l 1: "Table 2 resumes" → "Table 2 summarises"**

    AR: We replaced "resumes" by "summarizes". Thank you.

21. **RC: p 16, l 24: "punctual" → "point"**

    AR: We replaced "punctual" by "point". Thank you.

22. **RC: p 17, l 1: do you mean "lower fraction of diffuse radiation"?**

    AR: We changed "fraction" for "amount "

    AC: "higher amount of the diffuse radiation implies higher $GPP$."

23. **RC: p 17, l 18: "weighting" → "weighted"**

AR: Thank you.

24. **RC: p 17, l 25-28: not clear what you mean here, please rephrase**

AR: This has been modified

AC: The original sentence: 'Nevertheless, it is interesting to notice that the relative contribution of the diffuse to the total (diffuse + direct) aerosol effect on the $NEE$ (Equation 4) has a quite distinct behavior depending on the biome type and exponentially decay, or increase, with the AOD increase for all biomes, and C4 grass type, respectively.'

has been now replaced with

' Nevertheless, it is interesting to note that the impact of the aerosol influence on the relative contribution of the diffuse to the total (diffuse + direct) on the $NEE$ (Equation 4) has a different behavior depending on plant functional type, decaying exponentially as the AOD increases for all biomes, except for the C4 grass type.

25. **RC: p 18, l 2: please replace "it is fair to say" with a more scientific wording**

AR: We replaced. Thank you.

AC: "it is reasonable to say that the contribution..."

26. **RC: p 18, l 12-14: not sure what you mean here, please rephrase this last sentence of the paragraph**

AR: We rephrased it.

AC: The sentence: 'The difference between modeling and observational estimation for $NEE$ is likely to be within the yearly, and spatial variability of forest ecosystem physiology, which also includes disturbed areas and secondary forest.' has been changed to P16L26 and replaced with 'Table 3 shows that the

$NEE$ observed during the dry season at the Amazon forest and pasture biomes exhibit substantial site to site and interannual variability. Nevertheless, for each site, the 2010 model results are within the observed variability.'

27. **RC: p 19, l 19: remove "all" from "all together"**

    AR: This has been modified. Thank you.

28. **RC: p 27, Table 1 caption: "three-degree" → "third-degree"**

    AR: This has been changed. Thank you.

29. **RC: p 33, Fig 3b caption: please clarify what standard deviations are shown for the black and red lines (e.g. what values were used to derive them)**

    AR: The standard deviations were calculated using the mean of the 72 stations showed in Figure 3a with site locations represented in white asterisks.

    AC: The original sentence in Figure 3 caption :'The standard deviation of mean temperature from observation and from the model results are indicated by shaded gray and red bars, respectively.' has been replaced with:

    'The standard deviation (shaded gray) and the mean observed temperature values were calculated using measurements at the 72 observational stations. While the model standard deviation (red bars) and mean temperature were calculated using the model temperatures at the gridboxes corresponding to the locations of the 72 stations.

30. **RC: p 34, Fig 4: since this shows precipitation, I suggest to reverse the colour scale (as you did for Fig 5)**

    AR: This has been changed (see Figure 4 below)

31. **RC: p 36, Fig 6 caption: please clarify what exactly you mean by "fire product"**

AR: We changed the phrase.

AC: Changed: "Fire product derived from AVHRR measurements during September 2010"

To: "Burning points observed by the AVHRR sensor during September 2010."

32. **RC: p 37, Fig 7: please include some stats (here or in the text) for the scatter plots on the right. Also, the standard deviations are missing from the top scatter plots. I would also suggest to use a better colour scale for altitude to help visualising the results (at the moment the purple and pink are too similar – a more intuitive transition from low to high altitudes is preferable)**

AR: We changed (see Figure 7 below). The standard deviations that are not appearing is due to the fact that they are very small.

33. **RC: p 39, Fig 9: please use different colors for the model results (the light pink is almost invisible). Since you use UTC, it might help to show with dashed vertical lines where the local sunrise/sunset times are on the X-axis.**

AR: We used a darker color to represent the "NO-AER", so it should be visible now. We also included a bar with short wave radiation to indicate sunset and sunrise. (see Figure 9 below). Additionally, we included the bar with short wave radiation also in Figure 3b (see below).

34. **RC: p 40, Fig 10: why not showing the effect of the best simulation (DIR+DIF)?**

AR: The legend is wrong, this figure is really of DIR+DIF. Thank you.

35. **RC: p 44, Fig 14: please add a legend**

AR: The legend was included (see Figure 14 below).

36. **RC: p 45, Fig 15: please clarify what values are shown here (spatially and temporally)**

    AR: It is now indicated in the legend that the value is spatial over LBAR and temporal on September 2010.

    AC: ...during September 2010 (temporal) in the LBAR (spatial).

37. **RC: p 47, Fig 17: please add a legend**

    AR: The legend is now included (see Figure 17 below).

Please also note the supplement to this comment:
https://www.atmos-chem-phys-discuss.net/acp-2016-1147/acp-2016-1147-AC1-supplement.pdf

[Figure]

[Figure]

**Fig. 1.** Figure 3.

[Figure]

**Fig. 2.** Figure 4.

[Figure]

**Fig. 3.** Figure 7.

[Figure]

**Fig. 4.** Figure 9.

[Figure]

**Fig. 5.** Figure 14.

[Figure]

**Fig. 6.** Figure 17.

---

## Author Comment (AC2) · 1 Jul 2017

We thank the referee for his(er) insightful and very helpful comments, which contributed to improve the paper. The answers to his(er) questions and comments are below:

| **Legend:** | RC: Referee's Comment |
| --- | --- |
| | AR: Author's response |
| | AC: Author's changes in manuscript |

1. **RC: Unfortunately, the wording is often quite particular (shouldn't the title read '...in the Amazon region' or similar), despite the English language being overall comprehensible. Examples of such a particular wording which provides wrong spelling, twisted logic as well as unusual usage of words are P5L9 ": : :mixing ratios are diagnosed from the prognostic variables using the saturation mixing ratio with respect to liquid water", P12L6 ": : :fire emissions were not expected to contribute only minorly to $CO_2$ mixing ratios: : :", or P14L22 ": : : a high $GPP$ for C4 plants, but not high enough to compromise their photosynthesis process". A considerable language editing should therefore be carried out. This can be accompanied with extensive shortenings, particularly towards the end of the manuscript (i.e. P16-20).**

   AR: We changed the title to: 'Modelling the radiative effects of biomass burning aerosols on carbon fluxes in the Amazon region' and the document has been reviewed by a co-author who has English as his first language.

2. **RC: Furthermore, the paper needs more emphasize on the biosphere model. The reader only learns that the JULES model has been used and that it had been evaluated for sites in the Amazon before. What has not been explained in the methodology section is how the model considers direct and diffuse radiation for photosynthesis or how this response depends on plant functional type. It is also important to know how radiation and temperature changes influence simulated respiration (calculating a fixed or variable fraction of photosynthesis being lost as 'growth respiration', exponential temperature dependence on maintenance respiration, allocation shifts regarding exudation or fine root turnover changes the effect decomposition,: : :?).**

   AR: Following this suggestion, we explicitly included a section about JULES and BRAMS as part of section 2.1 and added more information on radiation interception and photosynthesis calculations in JULES, see below

AC: This is the new section on Jules that contains previous text and more information on radiation, photosynthesis and respiration.

Biosphere model: The Joint UK land simulator (JULES)

JULES simulates the exchange of carbon, momentum, and energy between the land surface and the atmosphere. Additionally, it represents subsurface hydrological processes, plants photosynthesis and respiration, and vegetation and soil dynamics (Best et al., 2011; Clark et al., 2011).

Atmospheric aerosols influence ecosystem functioning via effects on $GPP$ from changes in quality and quantity of radiation but also indirectly via temperature effects on $GPP$ but also on plant and heterotrophic respiration. The photosynthesis-radiation scheme, in JULES, accounts for the effects of diffuse radiation on canopy photosynthesis, by splitting direct and diffuse radiation and sunlit and shaded leaves at each canopy layer. Specifically, the multilayer radiation scheme includes an explicit calculation of absorption and scattering of the direct beam and the diffuse radiation fluxes in both visible and near-infrared wavebands, at each canopy layer, using the two-stream approach from Sellers (1985). Additionally, the attenuation of non-scattered incident direct beam radiation (sun flecks) is calculated using the approach by Dai et al. (2004). At each canopy layer, JULES estimates the fraction of absorbed direct and diffuse photosynthetic active radiation (PAR) thus providing a vertical profile of intercepted radiation fields which allows calculation of photosynthesis at each canopy level. At each canopy layer, the fraction of sunlit and shaded leaves is estimated as a function of the canopy beam radiation extinction coefficient (as explained in Clark et al 2011), and it is assumed that shaded leaves absorb only diffuse radiation and sunlit leaves absorb all types of radiation. Photosynthesis at each

canopy layer is then estimated as the sum of sunlit and shaded leaf photosynthesis weighed by their respective fraction. Total canopy photosynthesis is estimated as the sum of the leaf –level fluxes in each layer scaled by leaf area of each canopy layer. Temperature effects on photosynthesis are simulated in JULESs via biochemistry, leaf respiration and effects of vapor pressure deficit (VPD) on stomatal conductance in response to the temperature (see details in Clark et al. 2011). The temperature response of leaf respiration is linked to the temperature response of maximum carboxylation activity of Rubisco (Vcmax) in JULES, which is described by a peaked response function. The temperature response of remaining maintenance respiration components is simulated also using the leaf respiration temperature function. Growth respiration is estimated as a proportion of net primary productivity (NPP). Heterotrophic respiration is simulated either using a Q10 temperature function or a RothC temperature function (Jenkinson 1990 as described in Clark et al. 2011).

Evaluation of the skill of JULES in simulating $GPP$ under high direct and high diffuse radiation conditions has been tested against flux sites in the Amazon and in temperate forest sites where direct and diffuse radiation measurements are available. This is shown in Figure 2 of Rap et al. (2015) at Tapajos and French Guyana in the Amazon and at two temperate forest sites in Mercado et al. (2009) (Figure 1). Investigation of the response of photosynthesis to changes in direct and diffuse radiation across relevant plant functional types for the Amazon region is carried out within this study.

3. **RC:The depicted model properties (simplifications) should be used in the discussion to point out the appropriateness of the processes or the need for improvements. One of the reasons why the sensitivity of the model is important is that the importance of the direct and indirect aerosol effects might actually been less important than it looks like. I refer to chapter 3.2 where it is mentioned that the direct aerosol effect (by shading)**

**reaches -100 Wm-2 (Tapajos 80-123 Wm$^{-2}$), which comes along with a certain amount of cooling. This corresponds to about 460 umol m-2 s-1 global radiation or roughly speaking 230 umol m$^{-2}$ s$^{-1}$ PAR reduction. On the other hand, Fig. 12 shows that the increase of diffuse PAR due to the indirect aerosol effect is from app. 250 to 800 = 550 umol m2 s$^{-1}$. If direct and diffuse radiation are similarly effective in the model (please explore), the aerosol effect by shading should thus be about half the magnitude of the increase in diffuse radiation. Since it seems to be smaller, the cooling effect (part of direct aerosol effect) seems to compensate for the greater part of the shading. In my opinion this should be discussed in greater detail, using the sensitivity of the model against temperature changes for argumentation.**

AR: Under increased biomass burning aerosol concentrations, the maximum reduction in shortwave radiation due to aerosol loading ranges between 50 and 100 W m-2 (Fig 10b) and this corresponds to the maximum reduction of the air temperature near surface, which is of approximately 1 degree Celsius (Fig 10C) . At midday (1600 UTC), this reduction in shortwave radiation corresponds to less than 10% of the maximum radiation, which reaches approximately 900-1000 W m-2 in most places of the Amazon (Fig 10 a). Additionally, according to our DIR+DIF simulations, the highest value of the diffuse fraction attained in the studied region was 0.4. Based on these values, we conducted a sensitivity analysis using JULES in order to investigate the changes in $GPP$ driven by these changes in radiation, temperature and diffuse radiation (Fig S.4 of supplementary document). This sensitivity analysis shows that i) for a 10% decrease in shortwave radiation there are minimal changes in $GPP$ (Fig S.4a), ii) a change in temperature of one degree also did not imply major changes in the simulated $GPP$ (Fig S.4b), and iii) an increase in the diffuse fraction equivalent of 40% increased $GPP$ by 39% , 71%, 4%, and 72%, respectively, in forest, C3, C4 grasses, and cerrado (shrubs) vegetation (Fig S.4c). We conclude from this sensitivity test that in this

particular case, the effect of reduction of shortwave radiation and temperature due to the increase of the diffuse radiation had a small effect of on $GPP$.

4. **RC: P7L4: I don't understand what is meant by 'spin up artifacts'. Usually spin-ups are used to avoid artifacts originating from uncertain initial conditions.**

   AR: Thanks, we rephrased.

   AC: . The model simulations were initialized on 15 August 2010 00:00 UTC and conducted for 45 days. We discarded the first 15 days as spin-up, and restricted our analysis to the month of September to avoid model artifacts related to the initial conditions.

5. **RC: P7L7ff: From Fig. 12 it is apparent that diffuse PAR is about 250 umol m2-s-1 under conditions of AOD = 0 (clear sky conditions). I guess that this is about 5 percent of the total radiation even if AOD is actually 0. It seems likely that some clouds are even increasing this fraction. On the other hand the DIR-AER scenario seems to exclude this part of the radiation, which causes a bias that underestimates radiation and thus photosynthesis. Can you comment on this?**

   AR: As described in P5L25, the data presented in Figure 1 passed by a filter that removed the days with clouds, so when AOD=0, the parameter "d" of equation 1 gives the diffuse fraction due to the scattering by atmospheric gases, not clouds. The CARMA radiation parameterized only the direct component of the solar radiation. Thus, the solar radiation that reaches the surface (rshort) was divided into a direct ($rshort * (1 - D)$) and a diffuse component ($rshort * D$). In the DIR+DIF scenario the diffuse fraction (D) was obtained by equation 1 and sent to JULES, where the fraction of absorbed direct and diffuse radiation at each canopy layer is estimated On the other hand under the DIR-AER and NO-AER scenarios, a diffuse fraction of zero was prescribed, therefore JULES receives all incoming

radiation as direct radiation and zero diffuse radiation, i.e. this guarantees that there is no underestimation of the radiation.

6. **RC: P8L6ff: Equations 5 and 6 seem superfluous to me. A short description in the text should suffice**

AR: We agree.

AC.1: The lines P8L4:P8L9 were removed from the text (The contribution of the direct ... definitions in Eq. 2, 3, and 4, respectively.)

AC.2: P17L29:P17L35: The contribution of the diffuse radiation effect to $NEE$ ($\Delta NEE_{diff}/\Delta NEE_{tot}$) versus AOD, for each biome, is depicted in Figure 18 along with its fitting functions. Over forest, the percentage of the diffuse radiation effect on $CO_2$ uptake decreases exponentially ($[\Delta NEE_{diff}/\Delta NEE_{tot}]_{forest} \approx e^{-0.9AOD}, R^2 = 0.7$) from 100% to 50% with the increase of aerosol loading, reaching a balance of 50% - 50% between the diffuse and direct effect, for AOD above 0.5. For C3 grass and *cerrado*, as expected, the contribution of the diffuse radiation effects tends to zero with the increase of AOD ($[\Delta NEE_{diff}/\Delta NEE_{tot}]_{cerrado,C3} \approx 0.7e^{-4AOD}, R^2 = 0.7$). While for C4 grass type, the contribution of the diffuse radiation to $NEE$ exponentially increases with AOD ($[\Delta NEE_{diff}/\Delta NEE_{tot}]_{C4} \approx e^{AOD}, R^2 = 0.9$), the C4 photosynthetic pathway does not rapidly saturate with the amount of light received.

AC.3: P48: The contribution of the diffuse radiation effect to $NEE$ ($\Delta NEE_{diff}/\Delta NEE_{tot}$) as a function of AOD in the LBAR, but separated with different colors for different types of vegetation. The model data were filtered for cloudiness and precipitation. Additionally, only model points with the same soil water factor within all the three experiments, and soil moisture difference below 0.001 $m^3m^{-3}$ were included. The fitting functions of the $\Delta NEE_{diff}/\Delta NEE_{tot}$ versus AOD for each biome are also shown in the figure.

7. **RC: P8L26/28: Why are there two different algorithm numbers (3B42 and as**

**3B43)?**

AR: The correct is 3B42, 3B43 was a typing error. Thank you.

8. **RC: P10L22: What is meant by 'several precipitation systems'?**

AR: The phrase was modified to become clearer.

AC: However, one must take into account that the measurement stations are very scarce in this region, and for this reason, part of the precipitation occurred in the region may not have been computed in the monthly accumulated data from ground based measurements .

9. **RC: P10L27ff: The description of the soil moisture is a bit confusing. I would like to know how the soil is considered and initialized in the model (soil depth, number of layers, stratification of potential water content).**

AR: In P6L11:P6L113 we described that the model was initialized with the soil moisture estimation from the operational product developed by Gevaerd and Freitas (2006) and available at CPTEC/INPE. However, we in fact did not describe the soil depth, the number of layers and the soil type. In the new version, a more complete description was included in P6L11.

AC: Data from the RADAMBRASIL project (Rossato et al., 1998) was used for the soil type in Brazil and data from FAO (Zobler, 1999) was used outside Brazil. The model was run with seven soil levels: 0.10, 0.35, 1.0, 2.25, 4.25, 7.25 and 12.25 m below the surface. Soil moisture was initialized with...

* * *
[Figure]

**Fig. 1.** Figure 7.

**Supplement:**

**Modelling the radiative effects of smoke aerosols on carbon fluxes in the Amazon region**

Demerval S. Moreira[1,2], Karla M. Longo[3,a], Saulo R. Freitas[3,a], Marcia A. Yamasoe[4], Lina M. Mercado[5,6], Nilton E. Rosário[7], Emauel Gloor[8], Rosane S. M. Viana[9], John B. Miller[10], Luciana V. Gatti[11,12], Kenia T. Wiedemann[13], Lucas K. G. Domingues[11,12], and Caio C. S. Correia[11,12]

[1]Universidade Estadual Paulista (Unesp), Faculdade de Ciências, Bauru, SP, Brazil.
[2]Centro de Meteorologia de Bauru (IPMet), Bauru, SP, Brazil.
[3]Centro de Previsão de Tempo e Estudos Climáticos, Instituto Nacional de Pesquisas Espaciais (INPE), Cachoeira Paulista, SP, Brazil.
[4]Departamento de Ciências Atmosféricas do Institudo de Astronomia, Geofísica e Ciências Atmosféricas, Universidade de *São Paulo* (USP), *São Paulo*, SP, Brazil.
[5]Geography, College of Life and Environmental Sciences, University of Exeter, Exeter, UK.
[6]Centre for Ecology and Hydrology (CEH), Wallingford, UK.
[7]Universidade Federal de *São Paulo* (UNIFESP), Campus Diadema, Diadema, SP, Brasil.
[8]School of Geography, University of Leeds, Woodhouse Lane, Leeds, UK.
[9]Departamento de Matemática, Universidade Federal de Viçosa (UFV), Viçosa, MG, Brazil.
[10]Global Monitoring Division, Earth System Research Laboratory, National Oceanic and Atmospheric Administration (NOAA), Boulder, Colorado 80305, USA.
[11]Centro de Ciências do Sistema Terrestre, Instituto Nacional de Pesquisas Espaciais (INPE), São José dos Campos, SP, Brazil.
[12]Instituto de Pesquisas Energéticas e Nucleares (IPEN) – Comissão Nacional de Energia Nuclear (CNEN), São Paulo, Brazil.
[13]Department of Ecology and Evolutionary Biology, University of Arizona, Tucson, AZ, USA.
[a]Now at Universities Space Research Association/Goddard Earth Sciences Technology and Research (USRA/GESTAR) at Global Modeling and Assimilation Office, NASA Goddard Space Flight Center, Greenbelt, MD, USA.

*Correspondence to:* Demerval S. Moreira (demerval@fc.unesp.br)

**Supplementary Document**

[Figure]

**Figure S.1.** Input data used to run the JULES offline. Black curve represents the reference value (x = 0 in Figure S.2)

[Figure]

**Figure S.2.** Sensitivity test using JULES offline, running for a month (September 2010) and using the input data values shown in Figure S.1

[Figure]

**Figure S.3.** Similar to Figure S.2, but decreasing and increasing the original value shown in Figure S.1.

[Figure]

**Figure S.4.** Similar to Figure S.3, but with a zoom on the X-axis to better analyze the effects found in the simulations.

[Figure]

**Figure S.5.** Scatter plot of AOD at 550 nm from the model according DIR+DIF simulation and MODIS retrievals.

[Figure]

**Figure S.6.** Monthly mean wind at 10 meter at 00 UTC (black arrow), and 10 UTC (white arrow). The blue shaded is the Amazon and Tapajos rivers and the white contour shows the states border. The location of the km 67 tower is indicated with the red cross.

[Figure]

**Figure S.7.** a) Scatter plot of the number of fires detected in Brazil from 1999 to 2016 in September multiplied by 3 versus the total number considering the detection from January to December. The red cross is the year 2010. b) The mean annual cycle of the total number of fires for years 1999-2016 and respective standard deviation (blue) and 2010 only (red). Fire counting using NOAA-12 and AQUA MODIS as reference in 1999-2001 and 2002-2016, respectively. Data source: https://queimadas.dgi.inpe.br/queimadas.

[Figure]

**Figure S.8.** $NEE$ (μmolC m$^{-2}$ s$^{-1}$) versus AOD at 550 nm wavelength from the DIR+DIF experiment at the gridbox nearest to the Jaru site location (green) and measurements from Yamasoe at al. (2006) at the same location for 19 September to 15 November of 2002 (black).